# Reliable Graph Neural Networks via Robust Aggregation

**Simon Geisler**     **Daniel Zügner**     **Stephan Günnemann**
Department of Informatics
Technical University of Munich
{geisler, zuegnerd, guennemann}@in.tum.de

## Abstract

Perturbations targeting the graph structure have proven to be extremely effective in reducing the performance of Graph Neural Networks (GNNs), and traditional defenses such as adversarial training do not seem to be able to improve robustness. This work is motivated by the observation that adversarially injected edges effectively can be viewed as additional samples to a node's neighborhood aggregation function, which results in distorted aggregations accumulating over the layers. Conventional GNN aggregation functions, such as a sum or mean, can be distorted arbitrarily by a single outlier. We propose a robust aggregation function motivated by the field of robust statistics. Our approach exhibits the largest possible breakdown point of 0.5, which means that the bias of the aggregation is bounded as long as the fraction of adversarial edges of a node is less than 50%. Our novel aggregation function, Soft Medoid, is a fully differentiable generalization of the Medoid and therefore lends itself well for end-to-end deep learning. Equipping a GNN with our aggregation improves the robustness with respect to structure perturbations on Cora ML by a factor of 3 (and 5.5 on Citeseer) and by a factor of 8 for low-degree nodes.

## 1 Introduction

Learning on graph data has gained strong attention in recent years, specifically powered by the success of graph neural networks [29, 34]. Like for classic neural networks, (non-)robustness to adversarial perturbations has shown to be a critical issue for GNNs as well [18, 61]. In contrast to other application domains of deep learning, adversaries on graphs are especially challenging because not only the attributes might be perturbed, but also the discrete structure. Recently, many effective attacks on graph neural networks have been proposed [5, 18, 50, 54, 59, 61], and there is strong evidence that attacking the graph structure is more effective than attacking the attributes [52, 61].

While recent research suggests that effective defenses against attribute attacks can be found, e.g. robust training [59], defenses against structure attacks remain an unsolved topic [18, 54, 60]. Moreover, approaches such as [24, 52], solely focus on defending against specific attack characteristics. On the contrary, Carlini and Wagner [10] show that heuristic defenses often can be bypassed. Thus, we design our model without attack-specific assumptions.

Message passing is the core operation powering modern GNNs [27]. In the message passing steps, a node's embedding is updated by aggregating over its neighbors' embeddings. In this regard, adversarially inserted edges add additional data points to the aggregation and therefore perturb the output of the message passing step. Standard aggregation functions like a sum can be arbitrarily distorted by only a single outlier. Thus, we reason that on top of the usual (potentially non-robust) neural network components, GNNs introduce additional (typically non-robust) aggregations. Note

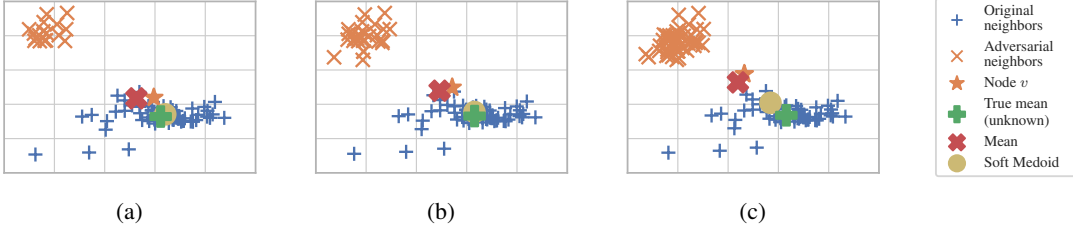

Figure 1: We show the output layer ($l = 2$) message passing step, i.e. the input of AGGREGATE$^{(l)}$, for adversarially added edges of an exemplary node $v$. The adversarial edges are obtained with a Nettack [61] evasion attack (at test time). For a two-dimensional visualization we used PCA on the weighted node embeddings $\mathbf{A}_{sw}\mathbf{h}_w^{(l-1)}\mathbf{W}^{(l)}$ of all edges $(s, w) \in \mathbf{A}$, but solely plot $v$'s neighborhood. We show the aggregation for 17, 29, and 50 perturbations in figure (a) to (c), respectively.

that many other countermeasures w.r.t. adversarial vulnerability are orthogonal to our approach and can be applied additionally.

We propose a novel robust aggregation function for GNNs to address this drawback. This aggregation function is novel in the context of deep learning. Our basic building block can be used within a large number of architectures by replacing the aggregation function with our proposed one. Our robust location estimator Soft Medoid is smooth and differentiable, which makes it well-suited for being used within a neural network, and it has the best possible breakdown point of 0.5. With an appropriate budget, the adversary can only perturb a subset of the aggregation inputs with the goal of crossing the decision boundary. As long as the adversary only controls the minority of inputs, our robust estimator comes with a bounded error regardless of the attack characteristics (i.e. no attack can distort the aggregation result arbitrarily).

Empirically, our method improves the robustness of its base architecture w.r.t. structural perturbations by up to 550% (relative), and outperforms previous state-of-the-art defenses. Moreover, we improve the robustness of the especially challenging to defend low degree nodes by a factor of 8.

## 2 Robust aggregation functions for graph neural networks

Throughout this work, we use the formulation in Eq. 1 (omitting the bias) for the message passing operation.

$$\mathbf{h}_v^{(l)} = \sigma^{(l)} \left( \text{AGGREGATE}^{(l)} \left\{ \left( \mathbf{A}_{vu}, \mathbf{h}_u^{(l-1)}\mathbf{W}^{(l)} \right), \forall\, u \in \mathcal{N}(v) \cup v \right\} \right) \tag{1}$$

$\mathbf{h}_v^{(l)}$ denotes the embedding of node $v$ in the $l$-th layer; $\mathbf{h}_v^{(0)}$ represents the (normalized) input features of node $v$. Further, $\mathbf{A}$ is the (potentially normalized) message passing matrix, $\mathbf{W}^{(l)}$ the parameter for the trainable linear transformation and $\sigma^{(l)}(\mathbf{z})$ the (non-linear) activation. $\mathcal{N}(v)$ is the set of neighbors of node $v$. GCN [34] instantiates Eq. 1 as $\mathbf{h}_v^{(l)} = \text{ReLU}(\text{SUM}\{(\mathbf{A}_{vu}\mathbf{h}_u^{(l-1)}\mathbf{W}^{(l)}), \forall\, u \in \mathcal{N}(v) \cup v\})$, where $\tilde{\mathbf{A}} = \hat{\mathbf{A}} + \mathbf{I}_N$, $\hat{\mathbf{D}}_{ii} = \sum_j \hat{\mathbf{A}}_{ij}$ and $\mathbf{A} = \hat{\mathbf{D}}^{1/2}\tilde{\mathbf{A}}\hat{\mathbf{D}}^{1/2}$ represents the normalization of the original adjacency matrix $\hat{\mathbf{A}}$. Common examples for AGGREGATE$^{(l)}$ are weighted mean[1] [1, 26, 34, 49], the max operation [29] or summation [55]. From a robust statistics point of view, a single perturbed embedding in $v$'s neighborhood suffices to arbitrarily deviate the resulting embedding $\mathbf{h}_v^{(l)}$. We hypothesize that the non-robustness of the aggregation function contributes to GNNs' non-robustness.

To back this hypothesis, we analyze the distortion of the neighborhood aggregation based on an exemplary message passing aggregation step in Fig. 1. The adversary inserts edges that result in a concentrated set of outliers. Only about 25% of outliers in the aggregation suffice to move the output outside of the convex hull of the clean data points. We see that a robust location estimator, such as the proposed Soft Medoid in Eq. 7, is much less affected by the outliers. Thus, we propose to use a *robust* aggregation function in the message passing operation Eq. 1.

Robustness of a location estimator has multiple facets. The breakdown point $\epsilon^*(t, \mathbf{X})$ (see Eq. 4) [22] measures the percentage of perturbed data points $\epsilon$ until the estimator $t$ can be arbitrarily distorted. It is well studied and has a probabilistic motivation for algebraically tailed distributions [39]. Complementary, the maxbias curve $B(\epsilon)$ (see Eq. 5) reports the maximum possible deviation of the location estimate between the clean and perturbed data w.r.t. the ratio of perturbed data [16]. Naturally, we desire a robust estimator to have a high breakdown point and low maxbias curve.

Measures such as the breakdown point are widely used as a proxy for the robustness of an estimator. While they analyze unbounded attacks, adversarially added edges in graph neural networks are, of course, not unbounded. However, for a strong/sufficient perturbation of the output, the attacker will likely perturb a neighborhood with nodes that have very different attributes/embeddings. Note that the magnitude of a structure perturbation is typically measured by the number of added or deleted edges (i.e. neighbors in Eq. 1). We investigate unbounded perturbations as a worst-case analysis and bounded attacks in our empirical evaluation. As we are going to see in Fig. 2, a robust estimator typically comes with a lower error for bounded perturbations as well.

Many such robust location estimators are computationally expensive or hard to implement in a vectorized fashion, and not continuously differentiable [19, 20, 23, 32, 39, 40, 46, 48]. In our experimentation, we found the M(R)CD estimator [8] and a differentiable dimension-wise median implementation (based on soft sorting [17]) computationally too demanding for the repeated message passing operation. Moreover, estimators for high dimensions [21] did not filter many adversarially added edges (perhaps the number of inputs to an aggregation in a GNN is too low).

We conclude that existing robust location estimators are ill-suited for use within a neural network, as fast computation and differentiability are crucial. Therefore we propose a novel robust and fully differentiable location estimator and base our aggregation function on the Medoid $t_{\text{Medoid}}(\mathbf{X}) = \arg\min_{\mathbf{y} \in \mathcal{X}} \sum_{j=1}^n \|\mathbf{x}_j - \mathbf{y}\|$, a multivariate generalization of the Median. In contrast to the $L_1$-Estimator $t_{L_1}(\mathbf{X}) = \arg\min_{\mathbf{y} \in \mathbb{R}^d} \sum_{j=1}^n \|\mathbf{x}_j - \mathbf{y}\|$, the Medoid constrains the domain of optimization from $\mathbf{y} \in \mathbb{R}^d$ to the input data points ($\mathbf{y} \in \mathcal{X}$). Throughout the paper, we denote the data matrix as $\mathbf{X}$ and its set representation with $\mathcal{X}$ interchangeably.

We propose a differentiable generalization of the Medoid replacing $\arg\min$ with a softmax to form a weighted average. That is,

$$t_{\text{Medoid}}(\mathbf{X}) = \arg\min_{\mathbf{y} \in \mathbf{X}} \sum_{j=1}^n \|\mathbf{x}_j - \mathbf{y}\| \approx \sum_{i=1}^n \hat{\mathbf{s}}_i \mathbf{x}_i = \hat{\mathbf{s}}^\top \mathbf{X} =: t_{\text{SM}}(\mathbf{X}) \,. \tag{2}$$

The weights $0 \leq \hat{\mathbf{s}}_i \leq 1$, $\sum_i \hat{\mathbf{s}}_i = 1$ are obtained via softmax of the data points' distances:

$$\hat{\mathbf{s}}_i = \frac{\exp\left(-\frac{1}{T} \sum_{j=1}^n \|\mathbf{x}_j - \mathbf{x}_i\|\right)}{\sum_{q=1}^n \exp\left(-\frac{1}{T} \sum_{j=1}^n \|\mathbf{x}_j - \mathbf{x}_q\|\right)} \,, \tag{3}$$

where $T$ is a temperature parameter controlling the steepness of the $\arg\min$ approximation. In this approximation, a point that has small distances to all other data points (i.e., a central data point) will have a large weight $\hat{\mathbf{s}}_i$, whereas remote points will have weights close to zero. For $T \to 0$ we recover the exact Medoid and for $T \to \infty$ the sample mean. Further, the range of the Soft Medoid is no longer limited to the data points themselves; it is now limited to the real numbers enclosed by the convex hull of the data points, i.e. $t_{\text{SM}}(\mathbf{X}) \in \mathcal{H}(\mathbf{X})$. Furthermore, due to the Euclidean distance, the (Soft) Medoid is *orthogonal equivariant* $t_{\text{SM}}(\mathbf{QX} + \mathbf{v}) = \mathbf{Q}\, t_{\text{SM}}(\mathbf{X}) + \mathbf{v}$, with the orthogonal matrix $\mathbf{Q}$ and the translation vector $\mathbf{v} \in \mathbb{R}^d$.

## 3 Robustness analysis

The (non-robust) sample mean and maximally robust Medoid are special cases of our smooth generalization of the Medoid (see Eq. 2), depending on the choice of the softmax temperature $T$. Naturally, this raises the question to what extent the Soft Medoid shares the robustness properties with the Medoid (or the non-robustness properties of the sample mean). In this section we show the non-obvious fact that regardless of the choice of $T \in [0, \infty)$ the Soft Medoid has an asymptotic breakdown point of $\epsilon^*(t_{\text{SM}}, \mathbf{X}) = 0.5$. As a corollary, the Soft Medoid comes with a guarantee on the embedding space. We conclude with a discussion of the influence of the temperature $T$. w.r.t. the maxbias curve.

The (finite-sample) breakdown point states the minimal fraction $\epsilon = m/n$ with $m$ perturbed examples, so that the result of the location estimator $t(\mathbf{X})$ can be arbitrarily placed [22]:

$$\epsilon^*(t, \mathbf{X}) = \min_{1 \leq m \leq n} \left\{ \frac{m}{n} : \sup_{\tilde{\mathbf{X}}_\epsilon} \|t(\mathbf{X}) - t(\tilde{\mathbf{X}}_\epsilon)\| = \infty \right\} \tag{4}$$

For this purpose, $\tilde{\mathbf{X}}_\epsilon$ denotes the perturbed data. To obtain $\tilde{\mathbf{X}}_\epsilon$ (equivalently $\tilde{\mathcal{X}}_\epsilon$) we may select and change up to $m$ (or an $\epsilon$ fraction of) data points of $\mathbf{x}_i \in \mathcal{X}$ and leave the rest as they are. Lopuhaä and Rousseeuw [39] show that for affine/orthogonal equivariant estimators such as the $L_1$-Estimator, the best possible breakdown point is $\epsilon^*(t_{L_1}, \mathbf{X}) = 0.5$. The sample mean, on the other side of the spectrum, has an asymptotic breakdown point of $\epsilon^*(t_\mu, \mathbf{X}) = 0$. A single perturbed sample is sufficient to introduce arbitrary deviations from the sample mean's true location estimate $t_\mu(\mathbf{X})$.

**Theorem 1** *Let $\mathcal{X} = \{\mathbf{x}_1, \ldots, \mathbf{x}_n\}$ be a collection of points in $\mathbb{R}^d$ with finite coordinates and temperature $T \in [0, \infty)$. Then the Soft Medoid location estimator (Eq. 2) has the finite sample breakdown point of $\epsilon^*(t_{SM}, \mathbf{X}) = 1/n \lfloor (n+1)/2 \rfloor$ (asymptotically $\lim_{n \to \infty} \epsilon^*(t_{SM}, \mathbf{X}) = 0.5$).*

Our analysis addresses the somewhat general question: How well do we need to approximate the Medoid or $L_1$-Estimator to maintain its robustness guarantees? Despite many approximate algorithms exits [9, 12, 13, 15, 25, 30, 33, 44, 45], we are the first to address this problem:

**Lemma 1** *Let $\mathcal{X} = \{\mathbf{x}_1, \ldots, \mathbf{x}_n\}$ be a collection of points in $\mathbb{R}^d$, which are (w.l.o.g.) centered such that $\hat{t}(\mathbf{X}) = 0$. Then, the (orthogonal equivariant) approximate Medoid or $L_1$-Estimator $\hat{t}$ has a breakdown point of $\epsilon^*(\hat{t}, \mathbf{X}) = 1/n \lfloor (n+1)/2 \rfloor$, if the following condition holds: $\lim_{p \to \infty} \hat{t}(\tilde{\mathbf{X}}_\epsilon)/p = 0$. Where $\tilde{\mathcal{X}}_\epsilon = \{\tilde{\mathbf{x}}_1, \ldots, \tilde{\mathbf{x}}_m, \mathbf{x}_{m+1}, \ldots, \mathbf{x}_n\}$ is obtained from $\mathcal{X}$ by replacing $m = \lfloor (n-1)/2 \rfloor$ arbitrary samples with a point mass on the first axis: $\tilde{\mathbf{x}}_i = [p \quad 0 \quad \cdots \quad 0]^\top, \forall i \in \{1, \ldots, m\}$.*

As a direct consequence of Lemma 1, it is not decisive how closely we approximate the true Medoid. The condition rather imposes an upper bound on the growth of the location estimator over the magnitude of the perturbation $p$. In addition to the formal proof in § A, we now present an illustrative proof sketch for a simplified scenario, which highlights why the Soft Medoid has such a strong guarantee regardless of $T \in [0, \infty)$ and omits the detour via Lemma 1.

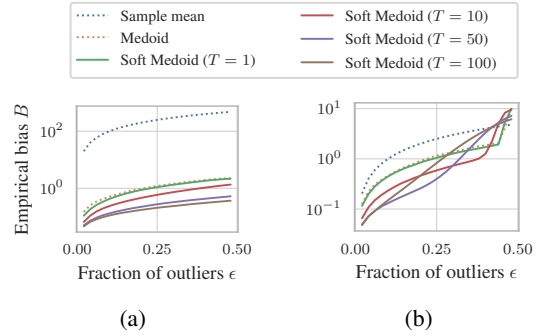

Figure 2: Empirical bias $B(\epsilon)$, for 50 samples from a centered ($t_{SM}(\mathbf{X}) = 0$) bivariate normal distribution. (a) shows the bias for a perturbation with norm 1000, and (b) 10.

*Proof Sketch* Due to the orthogonal equivariance we may choose $t_{SM}(\mathbf{X}) = 0$, without loss of generality. Let $\tilde{\mathcal{X}}_\epsilon$ be decomposable such that $\tilde{\mathcal{X}}_\epsilon = \tilde{\mathcal{X}}_\epsilon^{(\text{clean})} \cup \tilde{\mathcal{X}}_\epsilon^{(\text{pert.})}$. Clearly the worst-case perturbation is obtained when $\tilde{\mathcal{X}}_\epsilon^{(\text{pert.})}$ concentrates on a point mass [16]. Due to orthogonal equivariance we can, thus, pick $\tilde{\mathbf{x}}_i = [p \quad 0 \quad \cdots \quad 0]^\top, \forall \tilde{\mathbf{x}}_i \in \tilde{\mathcal{X}}_\epsilon^{(\text{pert.})}$ w.l.o.g. In the following, we analyze the special case where all clean data points are located in the origin $\mathbf{x}_i = 0, \forall \mathbf{x}_i \in \tilde{\mathcal{X}}_\epsilon^{(\text{clean})}$.

We now have to find the minimal fraction of outliers $\epsilon$ for which $\lim_{p \to \infty} \|t_{SM}(\tilde{\mathbf{X}}_\epsilon)\| < \infty$ does not hold anymore. Here, both terms in the equation of the Soft Medoid $t_{SM}(\tilde{\mathbf{X}}_\epsilon) = \hat{\mathbf{s}}^\top \tilde{\mathbf{X}}_\epsilon$ depend on $p$ and $\lim_{p \to \infty} t_{SM}(\tilde{\mathbf{X}}_\epsilon) = \lim_{p \to \infty} \hat{\mathbf{s}}^\top \tilde{\mathbf{X}}_\epsilon$ leads to the undefined case of $0 \cdot \infty$. However, because of $\lim_{x \to \infty} x e^{-x/a} = 0$ for $a \in [0, \infty)$, it turns out that we just have to analyze $\hat{\mathbf{s}}$ for $p \to \infty$. That is, if $\hat{\mathbf{s}}^{(\text{pert.})} \to 0$ the perturbed data have zero weight in the aggregation. We now relate the weight of any perturbed data point $\mathbf{s}^{(\text{pert.})}$ to the weight of any clean data point $\hat{\mathbf{s}}^{(\text{clean})}$:

$$\frac{\hat{\mathbf{s}}^{(\text{pert.})}}{\hat{\mathbf{s}}^{(\text{clean})}} = \frac{\exp\left\{ -\frac{1}{T} \sum_{\tilde{\mathbf{x}}_j \in \tilde{\mathcal{X}}_\epsilon} \|\tilde{\mathbf{x}}_j - \tilde{\mathbf{x}}^{(\text{pert.})}\| \right\}}{\exp\left\{ -\frac{1}{T} \sum_{\tilde{\mathbf{x}}_j \in \tilde{\mathcal{X}}_\epsilon} \|\tilde{\mathbf{x}}_j - \tilde{\mathbf{x}}^{(\text{clean})}\| \right\}} = \exp\left\{ -\frac{1}{T} \underbrace{\left[ \left( \sum_{\tilde{\mathbf{x}}_j \in \tilde{\mathcal{X}}_\epsilon^{(\text{clean})}} p \right) - \left( \sum_{\tilde{\mathbf{x}}_j \in \tilde{\mathcal{X}}_\epsilon^{(\text{pert.})}} p \right) \right]}_{\left( |\tilde{\mathcal{X}}_\epsilon^{(\text{clean})}| - |\tilde{\mathcal{X}}_\epsilon^{(\text{pert.})}| \right) \cdot p} \right\}$$

If we have more clean points than perturbed points ($|\tilde{\mathcal{X}}_\epsilon^{(\text{clean})}| > |\tilde{\mathcal{X}}_\epsilon^{(\text{pert.})}|$), then $\lim_{p \to \infty} \hat{\mathbf{s}}^{(\text{pert.})}/\hat{\mathbf{s}}^{(\text{clean})} = \exp(-\infty) = 0$. Note that $\hat{\mathbf{s}}^{(\text{pert.})}/\hat{\mathbf{s}}^{(\text{clean})} = 0$ can only be true if $\hat{\mathbf{s}}^{(\text{pert.})} = 0$. Hence, the norm of the Soft Medoid is finite when the perturbation $p$ approaches infinity iff $\epsilon < 0.5$. $\square$

For a corollary of Theorem 1, we formally introduce the (asymptotic) maxbias curve
$$B^*(\epsilon, t, \mathcal{D}_\mathcal{X}) = \sup_H \| t(\mathcal{D}_\mathcal{X}) - t\left((1-\epsilon)\mathcal{D}_\mathcal{X} + \epsilon H\right) \|, \tag{5}$$
with the data distribution $\mathcal{D}_\mathcal{X}$ and arbitrary distribution $H$ representing the perturbation. The maxbias curve models the maximum deviation between clean and perturbed estimate over different percentages of perturbations $\epsilon$. From Theorem 1 and the monotonicity of the maxbias curve, Corollary 1 follows.

**Corollary 1** *Let $\mathcal{X} = \{\mathbf{x_1}, \dots, \mathbf{x_n}\}$ be a collection of points in $\mathbb{R}^d$ with finite coordinates and the constant temperature $T \in [0, \infty)$. Then the Soft Medoid location estimator (Eq. 2) has a finite maxbias curve $B^*(\epsilon, t_{SM}, \mathcal{D}_\mathcal{X}) < \infty$ for $\epsilon < \epsilon^*(t_{SM}, \mathbf{X})$.*

There exists a finite upper bound on the maxbias, i.e. the maximum deviation $\|t_{\text{SM}}(\mathbf{X}) - t_{\text{SM}}(\tilde{\mathbf{X}}_\epsilon)\| < \infty$ between the estimate on the clean data $\mathbf{X}$ and perturbed data $\tilde{\mathbf{X}}_\epsilon$ is limited. Consequently, using the Soft-Medoid translates to robustness guarantees on the embedding space of each layer. However, deriving this upper bound analytically is out of scope for this work.

In Fig. 2, we give empirical results for a fixed point mass perturbation on the first axis over increasing values of $\epsilon$. Fig. 2 (a) shows that for high temperatures and distant perturbations our Soft Medoid achieves an even lower bias than the Medoid because it essentially averages the clean points. (b) shows that this comes with the risk of a higher bias for small perturbations and high $\epsilon$. However, in case the perturbation is close to the data points, the bias cannot be very high. In conclusion, in the context of a GNN and for an appropriate choice of $T$ as well as *bounded perturbations*, the Soft Medoid can help mitigate the effects of adversarially injected edges as long as $\epsilon$ is sufficiently small.

## 4 Instantiating the Soft Medoid for graph neural networks

Before we can show the effectiveness of our method, we need to discuss how we can use the proposed Soft Medoid in GNNs. Effectively, we have to extend Eq. 2 to the weighted case due to the weights in the respective message passing matrix $\mathbf{A}$:

$$\tilde{t}_{\text{WSM}}(\mathbf{X}, \mathbf{a}) = c\,(\mathbf{s} \circ \mathbf{a})^\top \mathbf{X} \quad (6) \qquad \mathbf{s}_i = \frac{\exp\left(-\frac{1}{T}\sum_{j=1}^n \mathbf{a}_j \|\mathbf{x}_j - \mathbf{x}_i\|\right)}{\sum_{q=1}^n \exp\left(-\frac{1}{T}\sum_{j=1}^n \mathbf{a}_j \|\mathbf{x}_j - \mathbf{x}_q\|\right)} \quad (7)$$

where $\mathbf{a}$ is a non-negative weight vector (e.g. the weights in a row of $\mathbf{A}$) and $c = (\sum_{j=1}^n \mathbf{a}_j)/(\sum_{j=1}^n \mathbf{s}_j \mathbf{a}_j)$.

Since the Soft Medoid interpolates between the Medoid and mean, we indeed have to adapt the location estimator at two places: The generalized definition of $\mathbf{s}$ handles the weighted Medoid, while $\mathbf{s} \circ \mathbf{a}$ resembles the weighted mean (note that for $T \to \infty$ all elements of $\mathbf{s}$ are equal, thus, using only $\mathbf{s}$ would result in an *unweighted* mean; $\mathbf{s} \circ \mathbf{a}$ makes it a weighted mean). The multiplication with $c$ simply ensures a proper normalization of $\tilde{t}_{\text{WSM}}$ like in a standard GNN.

Theorem 1 holds for the weighted case accordingly: Given a weight vector $\mathbf{a}$ with positive weights, the estimate $\tilde{t}_{\text{WSM}}$ cannot be arbitrarily perturbed if $\sum \mathbf{a}^{(\text{pert.})} < \sum \mathbf{a}^{(\text{clean})}$ is satisfied (see § A.4).

In Eq. 1 we plug in the newly derived Weighted Soft Medoid $\tilde{t}_{\text{WSM}}(\mathbf{X}, \mathbf{a})$ for the AGGREGATION. Thus, for node $v$ in layer $l$, $\mathbf{X}$ represents the stacked embeddings $\{\mathbf{h}_u^{(l-1)}\mathbf{W}^{(l)}, \forall u \in \mathcal{N}(v) \cup v\}$, and

Table 1: Average duration (time cost in ms) of one training epoch (over 200 epochs, preprocessing counts once). For the other defenses we used DeepRobust's implementation. We report "-" for an OOM. We used one 2.20 GHz core and one GeForce GTX 1080 Ti (11 Gb). For hyperparameters see § 5.

| GDC Prepr. | Cora ML [46] ✓ | | Citeseer [47] ✓ | | PubMed [46] ✓ | |
|---|---|---|---|---|---|---|
| SM GCN | 41.2 | 210.9 | 36.6 | 154.1 | 86.0 | 497.8 |
| SVD GCN | 119.4 | 120.8 | 66.3 | 67.3 | - | - |
| Jaccard GCN | 19.1 | 147.8 | 11.2 | 118.0 | 84.9 | 585.4 |
| RGCN | 8.7 | 7.5 | 6.3 | 9.3 | 135.5 | 136.6 |
| Vanilla GCN | 5.1 | 7.1 | 4.7 | 7.8 | 6.0 | 66.1 |
| Vanilla GAT | 15.2 | 65.6 | 11.8 | 53.3 | 46.4 | 270.8 |

$\mathbf{a}$ the weight vector consists of $\{\mathbf{A}_{vu}, \forall u \in \mathcal{N}(v) \cup v\}$. Hence, we can think about the terms before $\mathbf{X}$ in Eq. 6 as an input-dependent reweighting of the message passing matrix.

A sparse matrix implementation of the Weighted Soft Medoid has a time complexity of $O(n \sum_{v=1}^{n} (\deg(v) + 1)^2)$, with number of nodes $n$. Due to the power law distribution of many graphs, we will also have a few nodes with a very large degree. To circumvent this issue and to enable a fully vectorized implementation we propose to calculate the Weighted Soft Medoid for the embeddings of the $k$ neighbors with largest weight. This yields a time and space complexity of $O(nk^2)$ and for $k \ll n$ leads to a total worst-case complexity of $O(n)$. The time cost of the Soft Medoid (SM GCN) is comparable to the defenses SVD GCN and Jaccard GCN (see Table 1).

## 5 Experimental evaluation

In § 5.2, we discuss the influence of the temperature $T$. While our main focus is on evaluating *certifiable* robustness, we also analyze the empirical robustness via attacks (§ 5.3). In § 5.4 we present the main results and comparison to other defenses. We mainly highlight results on Cora ML and attacks jointly adding and deleting edges (for other datasets/attacks see § B). The source code is available at `https://www.daml.in.tum.de/reliable_gnn_via_robust_aggregation`.

### 5.1 Setup

**Architectures.** We compare our approach against the current state of the art defenses against structure attacks [24, 52, 58]. *SVD GCN* [24] performs a low-rank approximation of the adjacency matrix with a truncated SVD (the result is not sparse in general, we use rank 50), *Jaccard GCN* [52] use the Jaccard similarity on the attributes to filter dissimilar edges (we use a similarity threshold of 0.01), and the *RGCN* [58] models the graph convolution via a Gaussian distribution for absorbing the effects of adversarial changes. Further, we compare the robustness to the general-purpose GNNs Graph Attention Network (GAT) [49], Graph Diffusion Convolution (GDC) with a GCN architecture [36], and GCN [34]. As baselines of robust location estimators, we equip a GCN and a GDC with the dimension-wise Median and the Medoid. Note that because of their non-differentiability, only the gradient for the selected/central item is non-zero—similarly to, e.g., max-pooling on images.

**Datasets.** We evaluate these models on Cora ML [47], Citeseer [41], and PubMed [47] for semi-supervised node classification. § B.1 gives a summary of the size of the respective largest connected component, which we are using. None of the referenced attacks/defenses [5, 18, 24, 42, 50, 52, 54, 59, 61] uses a larger dataset. Note that our approach scales (runtime/space) with $\mathcal{O}(n)$ while SVD GCN has space complexity of $\mathcal{O}(n^2)$.

**Hyperparameters.** We use two-layer GNNs with default parameters, as suggested by the respective authors for all the models. We use the personalized PageRank version of GDC. For a fair comparison, we set the number of hidden units for all architectures to 64, the learning rate to $0.01$, weight decay to $5e-4$, and train for 3000 epochs with a patience of 300. For the architectures incorporating our Soft Medoid, we perform a grid search over different temperatures $T$ (for the range of the temperatures $T$ see Fig. 3). In case we are using GDC, we also test different values for the teleport probability $\alpha \in [0.05, 0.4]$. In the experiments on Cora ML and Citeseer we use $\alpha = 0.15$ as well as $k = 64$. We use $\alpha = 0.15$ as well as $k = 32$ in the PubMed experiments. For each approach and dataset, we rerun the experiment with three different seeds, use each 20 labels per class for training and validation, and report the one-sigma error of the mean.

**Robustness certificates.** To measure certifiable robustness, we use Randomized Smoothing [14, 37, 38] for GNNs [7]. Randomized smoothing is a probabilistic, black-box robustness certification technique that is applicable to any model. Following Bojchevski et al. [7] we create an ensemble of models $g(\mathbf{x})$ (aka the smooth classifier) that consists of the trained base classifier $f(\mathbf{x})$ with random inputs. We randomly perturbed the input via independent random flips of elements in the binary feature matrix and/or adjacency matrix. For adding an element we use the probability $p_a$ and for removing an element we use $p_d$.

We treat the prediction of a smooth classifier as *certifiably correct* if it is both correct and certifiably robust; i.e. the prediction does not change w.r.t. any of the considered perturbations/attacks. We refer to the ratio of certifiably correct predictions as the *certification ratio* $R(r_a, r_d)$ at addition radius $r_a$ and deletion radius $r_d$. For example, $R(r_a = 2, r_d = 0)$ denotes the ratio of nodes that are robust (and correct) under insertion of any two edges. Higher is better. We compare the robustness for three

different cases: (a) addition or deletion of edges, (b) only deletion, (c) only addition. For further details on randomized smoothing, we refer to § B.2.

Comparing all these certification ratios $R(r_a, r_d)$ at different radii is somewhat cumbersome and subjective. Therefore, we propose the accumulated certifications

$$\text{AC} = -R(0,0) + \sum_{r_a, r_d} R(r_a, r_d) \tag{8}$$

as a single measure that captures overall robustness. We decide to subtract $R(0,0)$, because it reflects the accuracy of the smooth classifier. This metric is related to the area underneath the bivariate certification ratio $R(r_a, r_d)$. Note that a more robust model has higher accumulated certifications.

To capture what certifiable radii one obtains for correct predictions, in Table 2, we additionally report the average certifiable radii $\bar{r}_a$ (and $\bar{r}_d$):

$$\bar{r}_a := \frac{1}{|C|} \sum_{i \in C} r_a^{\max}(i). \tag{9}$$

Here, $C$ denotes the set of all correctly predicted nodes and $r_a^{\max}(i)$ the maximum addition radius so that node $i$ can still be certified w.r.t. the smooth classifier $g(\mathbf{x}_i)$; analogously for $r_d$. The higher the better.

## 5.2 The temperature hyperparameter

Following up on the concluding statement of § 3, the temperature $T$ is a central hyperparameter for a GNN equipped with a Soft Medoid. Our best-performing setup is a GDC equipped with a Soft Medoid (see § 5.4). Consequently, we use this model for the analysis of the influence of $T$.

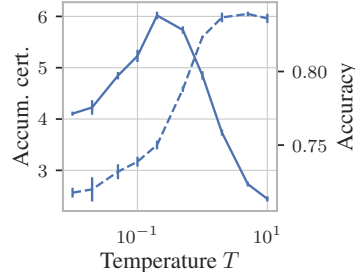

Fig. 3 illustrates this relationship for a wide range of $T$. Decreasing the temperature comes with increased robustness but at the cost of the accuracy. However, we cannot increase the robustness indefinitely and observe a maximum around $T = 0.2$. We hypothesize that this is because for too low values of $T$ the Soft Medoid ignores all but one input and for high temperatures $T$ we approach the non-robust sample mean. In reference to § 3, for the right temperature w.r.t. the

Figure 3: Influence of the temperature $T$ on the accumulated certifications (solid) and accuracy of the base classifier (dashed).

magnitude of perturbations, we essentially average over the clean data points. Depending on the requirements for the robustness accuracy trade-off, we conclude that the sweet spot is likely to be in the interval of $T \in [0.2, 1]$. With that in mind, we decide to report the reasonable trade-offs of $T = 1$, $T = 0.5$, and our most robust model ($T = 0.2$), for the experiments.

## 5.3 Empirical robustness

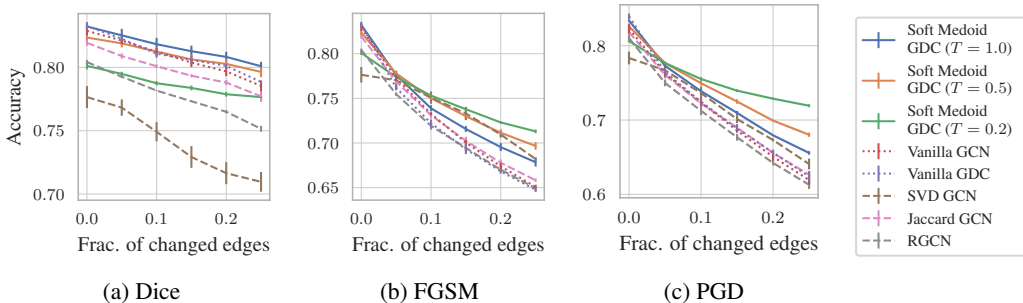

(a) Dice  (b) FGSM  (c) PGD

Figure 4: Accuracy for evasion (transfer) attacks on Cora ML.

The advantage of analyzing certifiable robustness is that it does not rely on specific attack approaches and the respective characteristic. However, the certificates we obtain are strictly speaking for the

resulting smooth classifier. As Cohen et al. [14] point out, only a base classifier that is robust w.r.t. these small perturbations can result in a robust smooth classifier. Still, for completeness, we report in Fig. 4 the (empirical) robustness of the base classifier, i.e. we measure how the accuracy drops when attacking the adjacency matrix. In such a scenario one has to refer to a specific attack approach. As shown, our approach outperforms all baselines with a significant margin for strong perturbations. That is, the accuracy stays high despite many perturbed edges. We report the perturbed accuracy for Dice [50], a FGSM-like [28] attack that greedily flips the element in **A** which contributes most to the test loss and Projected Gradient Descent (PGD) for $L_0$ perturbations [54]. For Nettack [61], Metattack [59], and the results on Citeseer see § B.3.

## 5.4 Certified robustness

In Table 2, we summarize the certified robustness of the experiments on Cora ML and selected experiments on Citeseer. For a complete comparison, we also report the accuracy of the base classifier. In § B.4, we report results on all three datasets with error estimates. Our Soft Medoid GDC architecture comes with a relative increase on the accumulated certifications of more than 200% w.r.t. adversarially added edges (most challenging case) for a wide range of baselines, alternative architectures, and defenses [24, 52, 58]. In the same scenario, on Citeseer we outperform the other baselines by a factor of 5.5. Moreover, our Soft Medoid GDC outperforms the "hard" Medoid as well as dimension-wise Median. As expected, increased robustness comes with the price of a slightly lower accuracy (compared to the best performing model which is substantially less robust).

Table 2: Accumulated certifications (first to third data column) and average certifiable radii (fourth and fifth data column) for the different architectures (top two highlighted). In the last column we list the clean accuracy of the base classifier (binary node attributes).

|  |  | Accum. certificates | | | Ave. cert. rad. | | Acc. |
|  |  | A.&d. | Add | Del. | Add | Del. |  |
|---|---|---|---|---|---|---|---|
| Cora ML [4] | Vanilla GCN | 1.84 | 0.21 | 4.42 | 0.25 | 5.37 | 0.823 |
|  | Vanilla GDC | 1.98 | 0.20 | 4.33 | 0.25 | 5.25 | 0.835 |
|  | Vanilla APPNP | 3.37 | 0.39 | 4.61 | 0.47 | 5.53 | **0.841** |
|  | Vanilla GAT | 1.26 | 0.07 | 4.03 | 0.09 | 5.02 | 0.806 |
|  | SVD GCN | 0.84 | 0.08 | 2.39 | 0.11 | 3.14 | 0.772 |
|  | Jaccard GCN | 0.86 | 0.01 | 4.39 | 0.02 | 5.43 | 0.775 |
|  | RGCN | 1.46 | 0.12 | 3.99 | 0.15 | 5.03 | 0.793 |
|  | SM GCN ($T = 50$) | 1.86 | 0.21 | 4.44 | 0.25 | 5.41 | 0.823 |
|  | Dimmedian GDC | 2.38 | 0.32 | 4.61 | 0.41 | 5.71 | 0.801 |
|  | Medoid GDC | 4.05 | 0.51 | 4.62 | 0.73 | 6.28 | 0.724 |
|  | SM GDC ($T = 1.0$) | 4.31 | 0.52 | 4.71 | 0.66 | 5.70 | 0.823 |
|  | SM GDC ($T = 0.5$) | 5.07 | 0.60 | 4.80 | 0.79 | 5.98 | 0.795 |
|  | SM GDC ($T = 0.2$) | **5.60** | **0.66** | **4.91** | **0.89** | **6.31** | 0.770 |
| Citeseer [4] | Vanilla GCN | 1.24 | 0.11 | 3.88 | 0.16 | 5.48 | 0.710 |
|  | SVD GCN | 0.52 | 0.00 | 2.12 | 0.00 | 3.25 | 0.639 |
|  | Jaccard GCN | 1.42 | 0.04 | 3.96 | 0.06 | 5.57 | 0.711 |
|  | RGCN | 1.12 | 0.09 | 3.89 | 0.12 | 5.44 | **0.719** |
|  | SM GDC ($T = 1.0$) | 2.67 | 0.32 | 4.12 | 0.45 | 5.77 | 0.711 |
|  | SM GDC ($T = 0.5$) | 3.62 | 0.48 | 4.22 | 0.69 | 5.94 | 0.709 |
|  | SM GDC ($T = 0.2$) | **4.69** | **0.60** | **4.44** | **0.89** | **6.32** | 0.702 |

**Graph diffusion.** Node degrees in real-world graphs typically follow a power-law distribution. Consequently, we must be able to deal with a large fraction of low degree nodes. To obtain more robust GNNs, methods that are increasing the degree of the nodes are an important ingredient for the success of our model. The GDC architecture [36] is one of the natural choices for smoothing the adjacency matrix because its low-pass filtering of the adjacency matrix leads to an increased number of non-zero weights.

To illustrate why the GDC architecture is well-suited for being equipped with the Soft Medoid, we plot the accumulated certifications over the degree in Fig. 5. We see that with increasing degree the Soft Medoid GCN can demonstrate its strengths. We hypothesize, given just a few data points (i.e. neighbors), it is challenging for a robust estimator to differentiate between clean samples and outliers. Moreover, just a few adversarially added edges suffice to exceed the breakdown point. Note, however, that GDC alone does not improve the robustness by much (see Fig. 6 and Fig. 5).

In conclusion of this discussion, the Soft Medoid and GDC synergize well and help to tackle the challenging problem of robustifying low degree nodes. In comparison to a GCN, with our approach, we can improve the robustness by up to eight times for low-degree nodes.

**Edge deletion.** For the case of edge deletion, a vanilla GCN performs already decently. This observation matches our experiments, where we found that with an identical budget it is more powerful to inject a few outliers than removing the same amount of "good" edges (in the sense of perturbing the message passing aggregation).

**Attributes.** We observed that increased robustness against structure attacks comes with a decreased robustness on attribute attacks (GCN as baseline). Since we do not focus attribute robustness, we

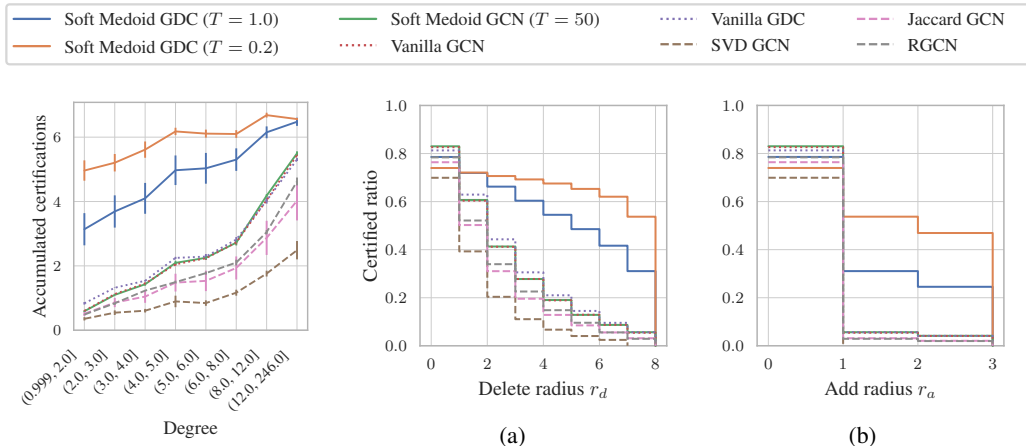

Figure 5: Accumulated certifications (see Eq. 8) over the degree (equal freq. binning).

Figure 6: (a) and (b) show the certification ratio over different radii for deletion $r_d$ and addition $r_a$. We compare our the Soft Medoid GDC against a GCN and the other defenses [24, 52, 58].

refer to § B.5 for further insights and, at the same time, we present a parametrization of our approach which comes with improved attribute robustness.

**Defenses.** Complementary to Table 2, in Fig. 6, we contrast the certification ratio for the Soft Medoid GDC to the state-of-the-art defenses [24, 52, 58] over different radii $r_d$ and $r_a$. Our model outperforms all of the tested state-of-the-art defenses by a large margin. All defenses [24, 52, 58] do not achieve high certification ratios. Thus, defenses designed for specific attacks cannot serve as general defenses against adversarial attacks. This highlights the need for certifiably robust models, as in general, we can make no a priori assumptions about adversarial attacks in the real world.

## 6 Related work

GNNs are an important class of deep neural networks, both from a scientific and application standpoint. Following the recent, trendsetting work in [29, 34], a vast number of approaches were proposed [1, 26, 35, 36, 49, 55]. A magnitude of adversarial attacks have been introduced [5, 18, 24, 42, 52, 54, 59, 61], pointing out their sensitivity regarding such attacks. Many of the proposed attacks directly propose an appropriate defense. We can classify the approaches into the categories of preprocessing [24, 52], robust training [54, 59], and modifications of the architecture [56, 58]. Perhaps the most similar approach, due to their statistical motivation, is RGCN [58].

An alternative direction to heuristic defenses is certification against small perturbations of the input [31, 51]. Some of these ideas were transferred to GNNs, recently [6, 60]. These certifications usually impose many restrictions regarding architectures or perturbations. In [7], randomized smoothing [14, 37, 38] was extended to GNNs for an empirical certification of arbitrary architectures.

Note that our reasoning about robust location estimators is orthogonal to the work of Xu et al. [55]. Their aim is to design aggregation functions that maximize the expressive power of a GNN. On the other hand, our goal is to design *robust* aggregation functions. Since the 1960s, the robust statistics community has been systematically studying such estimators in the presence of outliers [32, 48], and in recent years, research has also been drawn towards robust estimation in high dimensions [20].

## 7 Conclusion

We propose a robust aggregation function, Soft Medoid, for the internal use within GNNs. We show that the Soft Medoid—a fully differentiable generalization of the Medoid—comes with the best possible breakdown point of 0.5 and an upper bound of the error/bias of the internal aggregations. We outperform all baseline and the other defenses [24, 52, 58] w.r.t. robustness against structural perturbations by a relative margin of up to 450% and for low-degree edges even 700%.

## Broader Impact

This work is one step on the path towards the adversarial robustness of GNNs. Consequently, all potential applications of GNNs could benefit. These applications are computer vision, knowledge graphs, recommender systems, physics engines, and many more [53, 57]. Robust machine learning models certainly come with less opportunity of (fraudulent) manipulation. Robust models will enable the application of artificial intelligence (AI) for new use cases (e.g. safety-critical systems)—with all the related pros and cons. Perhaps, at some point, the discussion of risks and opportunities for AI [3, 11] and robust machine learning will converge. Focusing on the negative aspects of contemporary applications, robust GNNs might cause, e.g., an increased automation bias [43], or fewer loopholes e.g. in the surveillance implemented in authoritarian systems [2].

## Acknowledgments and Disclosure of Funding

This research was supported by the German Research Foundation, Emmy Noether grant GU 1409/2-1, the German Federal Ministry of Education and Research (BMBF), grant no. 01IS18036B, and the Helmholtz Association under the joint research school "Munich School for Data Science - MUDS." The authors of this work take full responsibilities for its content.

## Footnotes

[1]Technically we should call this operation weighted sum since the weights often do not sum up to 1. However, mean seems to be the widely accepted term (e.g. see [55]).

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
