[Supplementary Material]

# A  Proof of Soft Medoid breakdown point

Due to the extent of the proofs, we structure this section into three subsections. We start with a discussion of some preliminaries for the proofs. In § A.2, we build upon Lopuhaä and Rousseeuw [39]'s work to prove Lemma 1. In § A.3, we prove Theorem 1.

## A.1  Preliminaries

For this section, let $\tilde{\mathcal{X}}_\epsilon$ be decomposable such that $\tilde{\mathcal{X}}_\epsilon = \tilde{\mathcal{X}}_\epsilon^{(\text{clean})} \cup \tilde{\mathcal{X}}_\epsilon^{(\text{pert.})}$. Which is obtained from $\mathcal{X} = \{\mathbf{x}_1, \ldots, \mathbf{x}_n\}$, a collection of points in $\mathbb{R}^d$, by replacing up to $m$ points. The adversary can replace $m$ arbitrary points. For a concise notation we simply write that we replace the first $m$ values, but the points come with an arbitrary order beforehand.

Note that our analysis is easily extendable to two further possible definitions of $\tilde{\mathcal{X}}_\epsilon$: (a) the adversary adds a perturbation to the original values $\tilde{\mathcal{X}}_\epsilon = \{\mathbf{x}_1, \ldots, \mathbf{x}_m, \tilde{\mathbf{x}}_{m+1}, \ldots, \tilde{\mathbf{x}}_n\} = \{\mathbf{x}_1 + \mathbf{p}_1, \ldots, \mathbf{x}_m + \mathbf{p}_m, \mathbf{x}_{m+1}, \ldots, \mathbf{x}_n\}$ (b) the adversary adds $m$ perturbed samples to the dataset of $n$ clean samples. In case (a), just some values during the derivation change, but the results are essentially the same. For case (b), the number of samples is no longer $n$. Instead we have $m + n$ samples and need to adjust the equations accordingly. In this case the estimator is not broken down if $m < n$.

Recall that the Soft Medoid is orthogonal equivariant due to the Euclidean distance, i.e. $t_{\text{SM}}(\mathbf{Q}\mathbf{X} + \mathbf{v}) = \mathbf{Q}\, t_{\text{SM}}(\mathbf{X}) + \mathbf{v}$, with the orthogonal matrix $\mathbf{Q}$ and the translation vector $\mathbf{v} \in \mathbb{R}^d$. Lopuhaä and Rousseeuw [39] proved for their Lemma 2.1 that a orthogonal equivariant location estimator $t(\mathbf{X})$ has the same breakdown point, regardless of how the clean data points were rotated and/or translated, i.e. $\epsilon^*(t, \mathbf{A}\mathbf{X} + \mathbf{v}) = \epsilon^*(t, \mathbf{X})$. For any nonsingular orthogonal matrix $\mathbf{A}$ and translation vector $\mathbf{v}$. This is a very powerful statement and allows us to simplify our setup significantly. We may transform the clean datapoints such that $t_{\text{SM}}(\mathbf{X}) = 0$ and rotate them arbitrarily.

Croux et al. [16] derived a formula for the maxbias curve with Gaussian samples of an $L_1$ estimator and pointed out that the worst case perturbation is a point mass. To sketch why this must also hold for the Soft Medoid, we analyze the ratio of the softmax output of a perturbed sample $\hat{\mathbf{s}}_i$ over the softmax output of a clean sample $\hat{\mathbf{s}}_h$:

$$\frac{\hat{\mathbf{s}}_i}{\hat{\mathbf{s}}_h} = \frac{\exp\left\{-\frac{1}{T}\left[\overbrace{\sum_{q \in \tilde{\mathcal{X}}_\epsilon^{(\text{pert.})}} \|\tilde{\mathbf{x}}_q - \tilde{\mathbf{x}}_i\|}^{\alpha_1} + \overbrace{\sum_{o \in \tilde{\mathcal{X}}_\epsilon^{(\text{clean})}} \|\mathbf{x}_o - \tilde{\mathbf{x}}_i\|}^{\alpha_2}\right]\right\}}{\exp\left\{-\frac{1}{T}\left[\underbrace{\sum_{q \in \tilde{\mathcal{X}}_\epsilon^{(\text{pert.})}} \|\tilde{\mathbf{x}}_q - \mathbf{x}_h\|}_{\alpha_3} + \underbrace{\sum_{o \in \tilde{\mathcal{X}}_\epsilon^{(\text{clean})}} \|\mathbf{x}_o - \mathbf{x}_h\|}_{\alpha_4}\right]\right\}} \tag{10}$$

We see that we have four competing terms:

- $\alpha_1$ reflects some notion of variance of the perturbed samples (sum of distances from $\tilde{\mathbf{x}}_i$ to all other $\tilde{\mathcal{X}}_\epsilon^{(\text{pert.})}$)
- $\alpha_2$ is the sum of distances from the perturbed sample $\tilde{\mathbf{x}}_i$ to all clean samples
- $\alpha_3$ is the sum of distances from the clean sample $\mathbf{x}_h$ to all perturbed samples
- $\alpha_4$ reflects some notion of variance of the clean samples (sum of distances from $\mathbf{x}_h$ to all other $\tilde{\mathcal{X}}_\epsilon^{(\text{clean})}$)

Note that if our goal was to maximize the influence of a perturbed samples, we would need to minimize $\alpha_1$ and $\alpha_2$. Analogously, we need would need to maximize $\alpha_3$ and $\alpha_4$.

Hence, the bias benefits from a low variance of the perturbed samples $\alpha_1$ and high variance of the clean samples $\alpha_4$. The variance of the clean samples is something we typically cannot influence. $\alpha_1$ clearly shows that the perturbed samples do not deviate in any way that does not help to maximize $\hat{\mathbf{s}}_i/\hat{\mathbf{s}}_h$. We even obtain the smallest possible value $\alpha_1 = 0$, iff all perturbed samples coincide in one point. Moreover, $\alpha_2$ and $\alpha_3$ have a somewhat opposing objective. We can minimize $\alpha_2$, by locating the perturbed sample $\tilde{\mathbf{x}}_i$ close to the clean samples. However, in this case, the clean estimate cannot be perturbed much.

Based on all these considerations, it is clear that it does not make sense that we have e.g. two groups of perturbed data points moving in opposite directions. For a *worst case analysis* regarding perturbing the estimate towards infinity, we also must move *all* perturbed data points towards infinity. Hence, we can simply model the worst-case perturbation as a point mass. Further, due to the orthogonal equivariance, we may assume w.l.o.g, for example, that the perturbation is located on the first axis since we can arbitrarily rotate the data beforehand. Additionally to the consideration of the location, for a perturbation approaching infinity, we are going to see that the solution is not depending on the absolute coordinates of the clean data points $\mathcal{X}$

For solving several limits throughout our proofs, we make use of the following limit laws:

- Law of addition: $\lim_{x \to a} [f(x) + g(x)] = \lim_{x \to a} f(x) + \lim_{x \to a} g(x)$
- Law of multiplication: $\lim_{x \to a} [f(x)g(x)] = (\lim_{x \to a} f(x)) (\lim_{x \to a} g(x))$
- Law of division: $\lim_{x \to a} f(x)/g(x) = \lim_{x \to a} f(x)/\lim_{x \to a} g(x)$   (if $\lim_{x \to a} g(x) \neq 0$)
- Power law: $\lim_{x \to a} [f(x)]^b = [\lim_{x \to a} f(x)]^b$
- Composition law: $\lim_{x \to a} f(g(x)) = f(\lim_{x \to a} g(x))$   (if $f(x)$ is continuous)

Further, note that a limit of a vector-valued function is evaluated element-wise. With $\lim_{x \to \infty} f(x)$ we denote the limit towards positive infinity $x \to +\infty$.

## A.2   Proof of Lemma 1

*Proof* Lemma 1: Let $\mathcal{X} = \{\mathbf{x}_1, \ldots, \mathbf{x}_n\}$ be the $n$ clean data points. An adversary may replace an $\epsilon$-fraction ($\epsilon = m/n$) and we denote the resulting set as $\tilde{\mathcal{X}}_\epsilon = \tilde{\mathcal{X}}_\epsilon^{(\text{clean})} \cup \tilde{\mathcal{X}}_\epsilon^{(\text{pert.})}$. $\tilde{\mathcal{X}}_\epsilon^{(\text{pert.})} = \{\tilde{\mathbf{x}}_1, \ldots, \tilde{\mathbf{x}}_m\}$ are the $m$ replaced points and $\tilde{\mathcal{X}}_\epsilon^{(\text{clean})} \subset \mathcal{X}$ the $n - m$ remaining original data points. To ensure that $\epsilon < 0.5$, we may replace up to $m = \lfloor n-1/2 \rfloor$ data points. Note that until Eq. 13, this proof is closely aligned to the ideas of Lopuhaä and Rousseeuw [39].

We define $M = \max_{\mathbf{x}_i \in \mathcal{X}} \|\mathbf{x}_i\|$ and the ball $B(\mathbf{0}, 2M)$ with its center in the origin and radius $2M$. Moreover, we define the buffer $b = \inf_{\mathbf{v} \in B(\mathbf{0}, 2M)} \leq \|t(\tilde{\mathbf{X}}_\epsilon) - \mathbf{v}\|$. $b$ denotes the minimum distance between $t(\tilde{\mathbf{X}}_\epsilon)$ and $B(\mathbf{0}, 2M)$ and from its definition $\|t(\tilde{\mathbf{X}}_\epsilon)\| \leq b + 2M$ follows. As a consequence for all $m$ perturbed data points $\tilde{\mathbf{x}}_q \in \tilde{\mathcal{X}}_\epsilon^{(\text{pert.})}$ we have

$$\|\tilde{\mathbf{x}}_q - t(\tilde{\mathbf{X}}_\epsilon)\| \geq \|\tilde{\mathbf{x}}_q\| - \|t(\tilde{\mathbf{X}}_\epsilon)\| \geq \|\tilde{\mathbf{x}}_q\| - (b + 2M) \tag{11}$$

Suppose that $t(\tilde{\mathbf{X}}_\epsilon)$ and $B(\mathbf{0}, 2M)$ are far from each other, i.e. $b > 2Mm$. This definition seems to be arbitrary right now, but we are going to encounter exactly this term in Eq. 13 and this assumption will lead to a contradiction if the condition of Lemma 1 is satisfied. It is important to introduce this assumption right now, that we have for $\mathbf{x}_o \in \tilde{\mathcal{X}}_\epsilon^{(\text{clean})}$:

$$\|\mathbf{x}_c - t(\tilde{\mathbf{X}}_\epsilon)\| \geq M + b \geq \|\mathbf{x}_o\| + b \tag{12}$$

We now add Eq. 11 and Eq. 12 for all $q$ and $o$:

$$\sum_{\tilde{\mathbf{x}}_j \in \tilde{\mathcal{X}}_\epsilon} \|\tilde{\mathbf{x}}_j - t(\tilde{\mathbf{X}}_\epsilon)\| \geq \left( \sum_{\tilde{\mathbf{x}}_q \in \tilde{\mathcal{X}}_\epsilon^{(\text{pert.})}} \|\tilde{\mathbf{x}}_q\| - (b + 2M) \right) + \left( \sum_{\mathbf{x}_o \in \tilde{\mathcal{X}}_\epsilon^{(\text{clean})}} \|\mathbf{x}_o\| + b \right)$$

$$\geq \left( \sum_{\tilde{\mathbf{x}}_q \in \tilde{\mathcal{X}}_\epsilon^{(\text{pert.})}} \|\tilde{\mathbf{x}}_q\| \right) - mb - 2Mm + \left( \sum_{\mathbf{x}_o \in \tilde{\mathcal{X}}_\epsilon^{(\text{clean})}} \|\mathbf{x}_o\| \right) + (n - m)b$$

$$\geq \left( \sum_{\tilde{\mathbf{x}}_i \in \tilde{\mathcal{X}}_\epsilon} \|\tilde{\mathbf{x}}\| \right) + \underbrace{b - 2Mm}_{\substack{>0 \\ \text{(assumption)}}} > \sum_{\tilde{\mathbf{x}}_i \in \tilde{\mathcal{X}}_\epsilon} \|\tilde{\mathbf{x}}\|$$

$$\tag{13}$$

Note that $n - 2m = n - 2\lfloor n-1/2 \rfloor \geq 1$. More precisely, if $n$ is odd then $n - 2m = 1$, and if $n$ is even then $n - 2m = 2$.

Consequently, if we can show that

$$\sum_{\tilde{\mathbf{x}}_i \in \tilde{\mathcal{X}}_\epsilon} \|\tilde{\mathbf{x}}_i - t(\tilde{\mathbf{X}}_\epsilon)\| \leq \sum_{\tilde{\mathbf{x}}_i \in \tilde{\mathcal{X}}_\epsilon} \|\tilde{\mathbf{x}}\| \,, \tag{14}$$

or equivalently

$$\frac{\sum_{\tilde{\mathbf{x}}_i \in \tilde{\mathcal{X}}_\epsilon} \|\tilde{\mathbf{x}} - t(\tilde{\mathbf{X}}_\epsilon)\|}{\sum_{\tilde{\mathbf{x}}_i \in \tilde{\mathcal{X}}_\epsilon} \|\tilde{\mathbf{x}}\|} \leq 1 \,, \tag{15}$$

we have a contradiction and, hence, $b > 2Mm$ cannot be true. Similarly to [39], this leads us to the worst case guarantee of $\sup_{\tilde{\mathbf{X}}_\epsilon} \|t(\tilde{\mathbf{X}}_\epsilon) - t(\mathbf{X})\| \leq 2M(m+1) = 2\lfloor n+1/2 \rfloor M \leq (n+1) \max_{\mathbf{x}_i \in \mathbf{X}} \|\mathbf{x}_i\|$. Please acknowledge that this rather loose guarantee holds for asymptotically $\epsilon = 0.5$. This is the very worst case for which we can obtain a guarantee at all.

In the last step towards Lemma 1 (Eq. 16 and Eq. 17), we analyze the case of a point mass perturbation on the first axis such that $\tilde{\mathcal{X}}_\epsilon = \{\tilde{\mathbf{x}}_1, \ldots, \tilde{\mathbf{x}}_m, \mathbf{x}_{m+1}, \ldots, \mathbf{x}_n\}$ be the perturbed collection of points, with $m = \lfloor (n-1)/2 \rfloor$ perturbed points $\tilde{\mathbf{x}}_i = [p \quad 0 \quad \cdots \quad 0]^\top, \forall i \in \{1, \ldots, m\}$.

The nature of the breakdown point definition (see Eq. 4 in main part) requires us to show that the location estimate can approach infinity, once it has broken down. Hence, the factors need to grow indefinitely, which they only can if the perturbation $p$ approaches infinity. We are now going to analyze the resulting left side of Eq. 15:

$$
\begin{aligned}
&\lim_{p \to \infty} \frac{\sum_{\tilde{\mathbf{x}}_i \in \tilde{\mathcal{X}}_\epsilon} \|\tilde{\mathbf{x}} - t(\tilde{\mathbf{X}}_\epsilon)\|}{\sum_{\tilde{\mathbf{x}}_i \in \tilde{\mathcal{X}}_\epsilon} \|\tilde{\mathbf{x}}\|} \\
&= \lim_{p \to \infty} \frac{\sum_{\tilde{\mathbf{x}}_q \in \tilde{\mathcal{X}}_\epsilon^{(\mathrm{pert.})}} \|\tilde{\mathbf{x}}_q - t(\tilde{\mathbf{X}}_\epsilon)\| + \sum_{\mathbf{x}_o \in \tilde{\mathcal{X}}_\epsilon^{(\mathrm{clean})}} \|\mathbf{x}_o - t(\tilde{\mathbf{X}}_\epsilon)\|}{\sum_{\tilde{\mathbf{x}}_q \in \tilde{\mathcal{X}}_\epsilon^{(\mathrm{pert.})}} \|\tilde{\mathbf{x}}_q\| + \sum_{\mathbf{x}_o \in \tilde{\mathcal{X}}_\epsilon^{(\mathrm{clean})}} \|\mathbf{x}_o\|} \\
&= \lim_{p \to \infty} \frac{\frac{1}{p} \sum_{\tilde{\mathbf{x}}_q \in \tilde{\mathcal{X}}_\epsilon^{(\mathrm{pert.})}} \|\tilde{\mathbf{x}}_q - t(\tilde{\mathbf{X}}_\epsilon)\| + \sum_{\mathbf{x}_o \in \tilde{\mathcal{X}}_\epsilon^{(\mathrm{clean})}} \|\mathbf{x}_o - t(\tilde{\mathbf{X}}_\epsilon)\|}{\frac{1}{p} \sum_{\tilde{\mathbf{x}}_q \in \tilde{\mathcal{X}}_\epsilon^{(\mathrm{pert.})}} \|\tilde{\mathbf{x}}_q\| + \sum_{\mathbf{x}_o \in \tilde{\mathcal{X}}_\epsilon^{(\mathrm{clean})}} \|\mathbf{x}_o\|} \\
&= \lim_{p \to \infty} \frac{\sum_{\tilde{\mathbf{x}}_q \in \tilde{\mathcal{X}}_\epsilon^{(\mathrm{pert.})}} \|\frac{\tilde{\mathbf{x}}_q - t(\tilde{\mathbf{X}}_\epsilon)}{p}\| + \sum_{\mathbf{x}_o \in \tilde{\mathcal{X}}_\epsilon^{(\mathrm{clean})}} \|\frac{\mathbf{x}_o - t(\tilde{\mathbf{X}}_\epsilon)}{p}\|}{\sum_{\tilde{\mathbf{x}}_q \in \tilde{\mathcal{X}}_\epsilon^{(\mathrm{pert.})}} \|\frac{\tilde{\mathbf{x}}_q}{p}\| + \sum_{\mathbf{x}_o \in \tilde{\mathcal{X}}_\epsilon^{(\mathrm{clean})}} \|\frac{\mathbf{x}_o}{p}\|} \\
&= \frac{\sum_{\tilde{\mathbf{x}}_q \in \tilde{\mathcal{X}}_\epsilon^{(\mathrm{pert.})}} \lim_{p \to \infty} \|\frac{\tilde{\mathbf{x}}_q - t(\tilde{\mathbf{X}}_\epsilon)}{p}\| + \sum_{\mathbf{x}_o \in \tilde{\mathcal{X}}_\epsilon^{(\mathrm{clean})}} \lim_{p \to \infty} \|\frac{\mathbf{x}_o - t(\tilde{\mathbf{X}}_\epsilon)}{p}\|}{\sum_{\tilde{\mathbf{x}}_q \in \tilde{\mathcal{X}}_\epsilon^{(\mathrm{pert.})}} \lim_{p \to \infty} \|\frac{\tilde{\mathbf{x}}_q}{p}\| + \sum_{\mathbf{x}_o \in \tilde{\mathcal{X}}_\epsilon^{(\mathrm{clean})}} \lim_{p \to \infty} \|\frac{\mathbf{x}_o}{p}\|} \\
&= \frac{\sum_{\tilde{\mathbf{x}}_q \in \tilde{\mathcal{X}}_\epsilon^{(\mathrm{pert.})}} \lim_{p \to \infty} \sqrt{\left(\frac{p - t(\tilde{\mathbf{X}}_\epsilon)_1}{p}\right)^2 + \sum_{c=2}^{d}\left(\frac{t(\tilde{\mathbf{X}}_\epsilon)_c}{p}\right)^2} + \sum_{\mathbf{x}_o \in \tilde{\mathcal{X}}_\epsilon^{(\mathrm{clean})}} \lim_{p \to \infty} \sqrt{\sum_{c=1}^{d}\left(\frac{\mathbf{x}_{o,c} - t(\tilde{\mathbf{X}}_\epsilon)_c}{p}\right)^2}}{\sum_{\tilde{\mathbf{x}}_q \in \tilde{\mathcal{X}}_\epsilon^{(\mathrm{pert.})}} 1} \\
&= \frac{\sum_{\tilde{\mathbf{x}}_q \in \tilde{\mathcal{X}}_\epsilon^{(\mathrm{pert.})}} \sqrt{\left(1 - \lim_{p \to \infty}\frac{t(\tilde{\mathbf{X}}_\epsilon)_1}{p}\right)^2 + \sum_{c=2}^{d}\left(\lim_{p \to \infty}\frac{t(\tilde{\mathbf{X}}_\epsilon)_c}{p}\right)^2} + \sum_{\mathbf{x}_o \in \tilde{\mathcal{X}}_\epsilon^{(\mathrm{clean})}} \sqrt{\sum_{c=1}^{d}\left(\lim_{p \to \infty}\frac{t(\tilde{\mathbf{X}}_\epsilon)_c}{p}\right)^2}}{|\tilde{\mathcal{X}}_\epsilon^{(\mathrm{pert.})}|}
\end{aligned}
\tag{16}
$$

Since $t(\tilde{\mathbf{X}}_\epsilon)$ is a vector-valued function, we denote its $c$-th component with $t(\tilde{\mathbf{X}}_\epsilon)$ and, similarly, $\mathbf{x}_{o,c}$ stands for the $c$-th component of vector $\mathbf{x}_o$. Strictly speaking, the condition of Lemma 1 is not the only way to satisfy that Eq. 16 is $\leq 1$. In the following, we focus on the most relevant case, though.

If $\lim_{p\to\infty} t(\tilde{\mathbf{X}}_\epsilon)/p = \mathbf{0}$, Eq. 16 is $\leq 1$:

$$\lim_{p\to\infty} \frac{\sum_{\tilde{\mathbf{x}}_i\in\tilde{\mathcal{X}}_\epsilon} \|\tilde{\mathbf{x}} - t(\tilde{\mathbf{X}}_\epsilon)\|}{\sum_{\tilde{\mathbf{x}}_i\in\tilde{\mathcal{X}}_\epsilon} \|\tilde{\mathbf{x}}\|} = \frac{\sum_{\tilde{\mathbf{x}}_q\in\tilde{\mathcal{X}}_\epsilon^{(\mathrm{pert.})}} \sqrt{(1-0)^2 + \sum_{c=2}^{d}(0)^2} + \sum_{\mathbf{x}_o\in\tilde{\mathcal{X}}_\epsilon^{(\mathrm{clean})}} \sqrt{\sum_{c=1}^{d}(0)^2}}{|\tilde{\mathcal{X}}_\epsilon^{(\mathrm{pert.})}|} = \frac{|\tilde{\mathcal{X}}_\epsilon^{(\mathrm{pert.})}|}{|\tilde{\mathcal{X}}_\epsilon^{(\mathrm{pert.})}|} \leq 1 \tag{17}$$

Hence, the proof and Lemma 1 follows. $\square$

## A.3 Breakdown point of the Soft Medoid

*Proof* Theorem 1: Following up on Lemma 1, we need to show that $\lim_{p\to\infty} t_{\mathrm{SM}}(\tilde{\mathbf{X}}_\epsilon)/p = \lim_{p\to\infty} 1/p\, \hat{\mathbf{s}}^\top \tilde{\mathbf{X}}_\epsilon = \mathbf{0}$. We can expand the limit of the vector-matrix multiplication $1/p\, \hat{\mathbf{s}}^\top \tilde{\mathbf{X}}_\epsilon$ for its $c$-th component as (the result is going to be a vector):

$$\left(\lim_{p\to\infty} \frac{\hat{\mathbf{s}}^\top \tilde{\mathbf{X}}_\epsilon}{p}\right)_c = \lim_{p\to\infty} \sum_{i=1}^{n} \frac{\hat{\mathbf{s}}_i \tilde{\mathbf{x}}_{i,c}}{p} = \sum_{i=1}^{n} \left(\lim_{p\to\infty} \hat{\mathbf{s}}_i\right)\left(\lim_{p\to\infty} \frac{1}{p}\tilde{\mathbf{x}}_{i,c}\right) \tag{18}$$

From Eq. 18 it is clear that we can take the element-wise limit of $\lim_{p\to\infty} \hat{\mathbf{s}}^\top$ and $\lim_{p\to\infty} 1/p\,\tilde{\mathbf{X}}_\epsilon$, before the vector-matrix multiplication. The result is simply a sum of products of one element from the vector $\mathbf{s}$ and one of the matrix $1/p\,\tilde{\mathbf{X}}_\epsilon$. Hence, we are going to analyze $\lim_{p\to\infty} 1/p\, \hat{\mathbf{s}}^\top \tilde{\mathbf{X}}_\epsilon = \left(\lim_{p\to\infty} \hat{\mathbf{s}}^\top\right)\left(\lim_{p\to\infty} 1/p\,\tilde{\mathbf{X}}_\epsilon\right)$.

First, we analyze $\lim_{p\to\infty} \tilde{\mathbf{X}}_\epsilon/p$, i.e. the data matrix $\tilde{\mathbf{X}}_\epsilon$ multiplied by the scalar factor of $1/p$:

$$\lim_{p\to\infty} \frac{\tilde{\mathbf{X}}_\epsilon}{p} = \lim_{p\to\infty} \frac{1}{p} \begin{bmatrix} p & 0 & \cdots & 0 \\ \vdots & \vdots & & \vdots \\ p & 0 & \cdots & 0 \\ \mathbf{x}_{m+1,1} & \mathbf{x}_{m+1,2} & \cdots & \mathbf{x}_{m+1,d} \\ \vdots & \vdots & & \vdots \\ \mathbf{x}_{n,1} & \mathbf{x}_{n,2} & \cdots & \mathbf{x}_{n,d} \end{bmatrix} = \begin{bmatrix} 1 & 0 & \cdots & 0 \\ \vdots & \vdots & & \vdots \\ 1 & 0 & \cdots & 0 \\ 0 & 0 & \cdots & 0 \\ \vdots & \vdots & & \vdots \\ 0 & 0 & \cdots & 0 \end{bmatrix} \tag{19}$$

Second, since all weights in Eq. 19 are zero, but for the first component of the perturbed samples, we solely need to show that the softmax of the perturbed samples is 0, i.e. $\hat{\mathbf{s}}_q = 0$ for any $q \in \tilde{\mathcal{X}}_\epsilon^{(\mathrm{pert.})}$. It is possible to show this directly. However, we go a different path to keep this proof brief. That is, analogously to the proof sketch of § 3, the fraction of the softmax output of a perturbed sample $\hat{\mathbf{s}}_q$ over the softmax output of a clean sample $\hat{\mathbf{s}}_c = 0$ (for $c \in \tilde{\mathcal{X}}_\epsilon^{(\mathrm{clean})}$) for $p \to \infty$:

$$\lim_{p\to\infty} \frac{\hat{\mathbf{s}}_q}{\hat{\mathbf{s}}_o} = \lim_{p\to\infty} \frac{\exp\left\{-\frac{1}{T}\sum_{\tilde{\mathbf{x}}_j\in\tilde{\mathcal{X}}_\epsilon} \|\tilde{\mathbf{x}}_j - \tilde{\mathbf{x}}_q\|\right\}}{\exp\left\{-\frac{1}{T}\sum_{\tilde{\mathbf{x}}_j\in\tilde{\mathcal{X}}_\epsilon} \|\tilde{\mathbf{x}}_j - \mathbf{x}_o\|\right\}}$$

$$= \lim_{p\to\infty} \underbrace{\frac{1}{\exp\left\{-\frac{1}{T}\sum_{\mathbf{x}_i\in\tilde{\mathcal{X}}_\epsilon^{(\mathrm{clean})}} \|\mathbf{x}_i - \mathbf{x}_o\|\right\}}}_{\beta} \frac{\exp\left\{-\frac{1}{T}\sum_{\tilde{\mathbf{x}}_i\in\tilde{\mathcal{X}}_\epsilon^{(\mathrm{clean})}} \|\mathbf{x}_j - \tilde{\mathbf{x}}_q\|\right\}}{\exp\left\{-\frac{1}{T}\sum_{\tilde{\mathbf{x}}_i\in\tilde{\mathcal{X}}_\epsilon^{(\mathrm{pert.})}} \|\tilde{\mathbf{x}}_j - \mathbf{x}_o\|\right\}}$$

$$= \beta \lim_{p\to\infty} \frac{\exp\left\{-\frac{1}{T}\sum_{\tilde{\mathbf{x}}_i\in\tilde{\mathcal{X}}_\epsilon^{(\mathrm{clean})}} \|\mathbf{x}_j - \tilde{\mathbf{x}}_q\|\right\}}{\exp\left\{-\frac{1}{T}\sum_{\tilde{\mathbf{x}}_i\in\tilde{\mathcal{X}}_\epsilon^{(\mathrm{pert.})}} \|\tilde{\mathbf{x}}_j - \mathbf{x}_o\|\right\}} \tag{20}$$

$$= \beta \exp\left\{-\frac{1}{T}\left[\sum_{\tilde{\mathbf{x}}_i\in\tilde{\mathcal{X}}_\epsilon^{(\mathrm{clean})}} \lim_{p\to\infty} \|\mathbf{x}_j - \tilde{\mathbf{x}}_q\| - \sum_{\tilde{\mathbf{x}}_i\in\tilde{\mathcal{X}}_\epsilon^{(\mathrm{pert.})}} \lim_{p\to\infty} \|\tilde{\mathbf{x}}_j - \mathbf{x}_o\|\right]\right\}$$

$$\lim_{p \to \infty} \frac{\hat{\mathbf{s}}_q}{\hat{\mathbf{s}}_o} = \beta \exp \left\{ -\frac{1}{T} \frac{\sum_{\tilde{\mathbf{x}}_i \in \tilde{\mathcal{X}}_\epsilon^{(\text{clean})}} \lim_{p \to \infty} \|\frac{\mathbf{x}_j - \tilde{\mathbf{x}}_q}{p}\| - \sum_{\tilde{\mathbf{x}}_i \in \tilde{\mathcal{X}}_\epsilon^{(\text{pert.})}} \lim_{p \to \infty} \|\frac{\tilde{\mathbf{x}}_j - \mathbf{x}_o}{p}\|}{\lim_{p \to \infty} \frac{1}{p}} \right\}$$

$$= \beta \exp \left\{ -\frac{1}{T} \frac{\sum_{\tilde{\mathbf{x}}_i \in \tilde{\mathcal{X}}_\epsilon^{(\text{clean})}} 1 - \sum_{\tilde{\mathbf{x}}_i \in \tilde{\mathcal{X}}_\epsilon^{(\text{pert.})}} 1}{\frac{1}{\infty}} \right\}$$

$$= \beta \exp \left\{ -\frac{1}{T} \frac{\overbrace{|\tilde{\mathcal{X}}_\epsilon^{(\text{clean})}| - |\tilde{\mathcal{X}}_\epsilon^{(\text{pert.})}|}^{>0}}{\frac{1}{\infty}} \right\} \tag{21}$$

$$= \beta \exp \{-\infty\}$$

$$= 0$$

Due to the range of the softmax $\hat{\mathbf{s}}_i \in [0,1], \forall i \in \{1, \ldots, n\}$, a division by $\infty$ in $\hat{\mathbf{s}}_q/\hat{\mathbf{s}}_o$ is not possible. It follows that $\hat{\mathbf{s}}_q/\hat{\mathbf{s}}_o = 0$ iff the numerator $\hat{\mathbf{s}}_q = 0$. Consequently, $\hat{\mathbf{s}}_q = 0, \forall q \in \tilde{\mathcal{X}}_\epsilon^{(\text{pert.})}$ holds. This means that the weights for the perturbed samples approaches zero if $p \to \infty$.

In conclusion, the location estimate cannot approach infinity as long as the clean data points are finite and $|\tilde{\mathcal{X}}_\epsilon^{(\text{clean})}| > |\tilde{\mathcal{X}}_\epsilon^{(\text{pert.})}|$, independent of the choice of $T \in [0, \infty)$. $\square$

### A.4 Robustness of the Weighted Soft Medoid

While the Weighted Soft Medoid's breakdown point holds for any positive weight vector, we present here quickly the results for the case of rational weights. In the context of a computer program, this is true anyways for numbers represented with finite precision. We only need to find the greatest common divisor of the weight vector $\gcd(\mathbf{a}) = \gcd([\mathbf{a}_1 \quad \ldots \quad \mathbf{a}_n])$.

*Proof* weighted version of Theorem 1: We can transform $\tilde{\mathcal{X}}_\epsilon = \{\tilde{\mathbf{x}}_1, \ldots, \tilde{\mathbf{x}}_m, \mathbf{x}_{m+1}, \ldots, \mathbf{x}_n\}$ with the according weight vector $\mathbf{a} = [\mathbf{a}_1 \quad \ldots \quad \mathbf{a}_n]$ to its unweighted equivalent by this simple procedure: (1) we calculate $\mathbf{a}_j/\gcd(\mathbf{a}) = w_j, j\forall\{1, \ldots, n\}$ and (2) we duplicate $\tilde{\mathbf{x}}_j$ exactly $w_j$ times and collect the results in the multiset $\underline{\tilde{\mathcal{X}}}_\epsilon$. Thereafter, we simply apply the unweighted Soft Medoid on the multiset $\underline{\tilde{\mathcal{X}}}_\epsilon$. It is apparent, that as long as

$$\sum_{q=1}^{m} \mathbf{a}_q < \sum_{o=m+1}^{n} \mathbf{a}_o \tag{22}$$

holds, there are less perturbed examples than clean examples in $\underline{\tilde{\mathcal{X}}}_\epsilon$ and, hence, the estimator cannot be broken down. $\square$

Note that the normalization etc. in Eq. 6 (§ 4) does not affect the breakdown point of the estimator. In Eq. 21, we show that the softmax weights for the perturbed samples approaches zero if $p \to \infty$. These zero weights are still zero after the normalization; hence, the normalization does not influence the breakdown point.

# B  Detailed experimental results

In this section, we present further and more detailed experimental results. We start with the dataset statistics in § B.1. Thereafter, in § B.2, we describe the setup regarding randomized smoothing in more detail. § B.4 summarizes the complete results of our experiments. In § B.5, we discuss the trade-off between robustness w.r.t. structural and attribute robustness. We conclude this section with further plots of the certification ratio over different radii.

## B.1  Dataset statistics

For our experiments we use the largest connected component of the very common datasets summarized in Table 3. In these citation graphs, the nodes of the graph represent publications and the edges citations. The node features are the one-hot encoding of the bag of words of the respective abstract. The classes of the semi-supervised prediction task represent different categories of the publications.

Table 3: Statistics of the largest connected component of the used datasets.

|  | #Nodes $n$ | #Edges $e$ | #Features $d$ |
|---|---|---|---|
| **Cora ML [4]** | 2,810 | 15,962 | 2,879 |
| **Citeseer [41]** | 2,110 | 7,336 | 3,703 |
| **PubMed [47]** | 19,717 | 88,648 | 500 |

## B.2  Randomized Smoothing

In randomized smoothing, a deterministic or random base classifier $f : \mathcal{X} \to \mathcal{Y}$, that is a function from $\mathbb{R}^d$ to the classes in $\mathcal{Y}$, is extended to a smoothed classifier such that:

$$g(\mathbf{x}) = \arg \max_{c \in \mathcal{Y}} \mathbb{P}(f(\phi(\mathbf{x})) = c) \tag{23}$$

Where $\phi(\mathbf{x})$ denotes some randomization around $\mathbf{x}$. In the general case, there is not much hope to obtain the ensemble in closed form solution. Hence, in a Monte Carlo sampling setting, $f(\phi(\mathbf{x}))$ is invoked multiple times. Then, the classification of the smooth ensemble is obtained via a majority vote among the different random inputs and the relative frequencies reflect class probabilities. Cohen et al. [14] showed that for Gaussian noise one can obtain the certifiable $L_2$-ball radius depending on the difference between the most likely $p_A$ and second most likely class $p_B$. A certification of a radius $r$ according to the $L_2$-ball means that with high probability $(1 - \alpha_{\text{smoothing}})$ the most likely class of the smooth classifier does not change if the perturbation of the input $\delta$ is less or equal to the certified radius: $\|\delta\| \leq r$. To certify a large radius, we need (a) large difference $p_A - p_B$, (b) a strong noise (e.g. high variance), and/or (c) many Monte Carlos samples of $f(\phi(\mathbf{x}))$.

However, this definition of a certifiable radius does not reflect if the prediction was correct in the first place. Similarly to Cohen et al. [14], it makes sense to introduce a metric that combines correct prediction and certifiable robustness. This is why we report this conjunction throughout our experiments (see § 5.1).

For the graph structure, we do have to deal with discrete values. Thus, applying Gaussian noise, as in [14], is not a good choice. Bojchevski et al. [7] extended the framework of randomized smoothing to discrete variables in such a way that it is suitable for GNNs—considering the sparsity. For this purpose, they use a independent Bernoulli random variables that depend on the original data as randomization scheme $\phi(\mathbf{x})$ and distinguish between a probability for deleting a binary feature or edge $p_-$, as well as for adding a binary feature or edge $p_+$. Lastly, we can obtain certificates at different radii for deletion $r_d$ and addition $r_a$ on the $L_0$-ball. Note that for real-world graphs we have much fewer edges than the nodes squared, which results in a sparse adjacency matrix $\mathbf{A}$ (most of the values are zero). If we used the same probability for adding and deleting edges, we would just delete a few edges but add comparatively many edges.

Cohen et al. [14] argue that only a base classifier that is robust against the small perturbations $\phi(\mathbf{x})$ can result in a certifiably robust smooth classifier. This is why we expect a base classifier that can be certified at a high radius to be more robust. For the smoothing we use the addition probability $p_+ = 0.001$ and deletion probability $p_- = 0.4$, as suggested by [7]. To obtain the certificates, we use a significance level of $\alpha_{\text{smoothing}} = 0.05$ and perform 10,000 forward passes.

## B.3 Empirical robustness

For the empirical robustness (see § 5.3), we use a surrogate GCN to perform the respective attack and adapt the adjacency matrix. We train the other models on the clean graph and only use the perturbed adjacency matrix for the prediction (evasion attack). Using the surrogate GCN comes with the main benefit that we do not evaluate how well a model might "obfuscate" the gradient towards the adjacency matrix. Moreover, every model has to face exactly the same changed edges.

In Fig. 7 we present the results on Citeseer in addition to Fig. 4 of the main part. We see that our Soft Medoid outperforms the other approaches significantly for strong perturbations. In Table 4 we present the results on Nettack, where we outperform the other approaches as well. SVD GCN is the only exception and performs on par with our Soft Medoid GDC. Note that SVD GCN is specifically designed for Nettack. Last, Table 5 complements Fig. 4 and Fig. 7 with se-

Table 4: Targeted attack in the same setup as the evasion Nettack attack [61]. We report the average margin and the failure rate of the attack (higher is better).

| | | Nettack | |
| | | Margin | Fail. r. |
|---|---|---|---|
| Cora ML [4] | Vanilla GCN | -0.41 ± 0.05 | 0.18 ± 0.06 |
| | Vanilla GDC | -0.48 ± 0.12 | 0.15 ± 0.10 |
| | SVD GCN | 0.21 ± 0.06 | **0.64 ± 0.05** |
| | Jaccard GCN | -0.46 ± 0.13 | 0.26 ± 0.07 |
| | RGCN | 0.00 ± 0.01 | 0.35 ± 0.01 |
| | SM GDC ($T = 1.0$) | 0.09 ± 0.03 | 0.54 ± 0.02 |
| | SM GDC ($T = 0.5$) | 0.11 ± 0.07 | 0.53 ± 0.08 |
| | SM GDC ($T = 0.2$) | **0.24 ± 0.06** | 0.62 ± 0.04 |
| Citeseer [41] | Vanilla GCN | -0.56 ± 0.04 | 0.03 ± 0.00 |
| | Vanilla GDC | -0.51 ± 0.02 | 0.05 ± 0.02 |
| | SVD GCN | -0.00 ± 0.11 | 0.51 ± 0.09 |
| | Jaccard GCN | -0.43 ± 0.07 | 0.17 ± 0.07 |
| | RGCN | -0.05 ± 0.03 | 0.41 ± 0.02 |
| | SM GDC ($T = 1.0$) | -0.08 ± 0.04 | 0.36 ± 0.04 |
| | SM GDC ($T = 0.5$) | 0.10 ± 0.07 | 0.51 ± 0.07 |
| | SM GDC ($T = 0.2$) | **0.31 ± 0.04** | **0.65 ± 0.05** |

lected numerical results including the standard error of the mean. Furthermore, the last two columns contain the results on the poisoning attack Metattack [59]. In conclusion, we see that our Soft Medoid GDC performs decently over a wide range of attacks. Having the results on certifiable robustness in mind (see § 5.4), this comes at no surprise since certifiable robustness is an attack-agnostic measure of robustness.

Figure 7: Accuracy for evasion (transfer) attacks on Citeseer.

Table 5: Perturbed accuracy for the global attacks on Cora ML and Citeseer. Here $\epsilon$ denotes the fraction of edges perturbed (relative to the clean graph).

| | Attack | Dice | | FGSM | | PGD | | Metattack | |
|---|---|---|---|---|---|---|---|---|---|
| | Frac. pert. edges $\epsilon$ | 0.10 | 0.25 | 0.10 | 0.25 | 0.10 | 0.25 | 0.10 | 0.25 |
| Cora ML [4] | Vanilla GCN | 0.812 ± 0.003 | 0.785 ± 0.004 | 0.732 ± 0.005 | 0.655 ± 0.003 | 0.724 ± 0.006 | 0.619 ± 0.006 | 0.555 ± 0.022 | 0.383 ± 0.012 |
| | Vanilla GDC | 0.811 ± 0.003 | 0.789 ± 0.001 | 0.733 ± 0.004 | 0.657 ± 0.003 | 0.725 ± 0.004 | 0.624 ± 0.007 | 0.547 ± 0.021 | 0.342 ± 0.012 |
| | SVD GCN | 0.749 ± 0.008 | 0.710 ± 0.008 | 0.750 ± 0.007 | 0.677 ± 0.005 | 0.736 ± 0.009 | 0.641 ± 0.007 | **0.699 ± 0.021** | **0.496 ± 0.011** |
| | Jaccard GCN | 0.801 ± 0.002 | 0.777 ± 0.004 | 0.733 ± 0.002 | 0.662 ± 0.001 | 0.722 ± 0.003 | 0.626 ± 0.005 | 0.584 ± 0.023 | 0.415 ± 0.007 |
| | RGCN | 0.782 ± 0.000 | 0.752 ± 0.002 | 0.721 ± 0.004 | 0.647 ± 0.004 | 0.712 ± 0.006 | 0.613 ± 0.006 | 0.591 ± 0.028 | 0.359 ± 0.012 |
| | SM GDC ($T = 1.0$) | **0.818 ± 0.005** | **0.801 ± 0.003** | 0.743 ± 0.001 | 0.679 ± 0.003 | 0.739 ± 0.002 | 0.656 ± 0.002 | 0.608 ± 0.028 | 0.433 ± 0.013 |
| | SM GDC ($T = 0.5$) | 0.813 ± 0.003 | 0.796 ± 0.004 | **0.751 ± 0.001** | 0.693 ± 0.002 | 0.750 ± 0.003 | 0.680 ± 0.003 | 0.626 ± 0.024 | 0.459 ± 0.024 |
| | SM GDC ($T = 0.2$) | 0.787 ± 0.002 | 0.777 ± 0.004 | 0.749 ± 0.002 | **0.702 ± 0.002** | **0.755 ± 0.002** | **0.719 ± 0.002** | 0.641 ± 0.026 | 0.474 ± 0.029 |
| Citeseer [41] | Vanilla GCN | 0.696 ± 0.016 | 0.678 ± 0.014 | 0.642 ± 0.014 | 0.570 ± 0.022 | 0.636 ± 0.009 | 0.556 ± 0.013 | 0.587 ± 0.021 | 0.439 ± 0.035 |
| | Vanilla GDC | 0.686 ± 0.010 | 0.666 ± 0.009 | 0.635 ± 0.011 | 0.563 ± 0.022 | 0.622 ± 0.012 | 0.548 ± 0.017 | 0.598 ± 0.017 | 0.450 ± 0.024 |
| | SVD GCN | 0.622 ± 0.016 | 0.582 ± 0.021 | 0.625 ± 0.016 | 0.566 ± 0.022 | 0.604 ± 0.015 | 0.545 ± 0.024 | **0.631 ± 0.019** | **0.531 ± 0.044** |
| | Jaccard GCN | 0.700 ± 0.017 | 0.683 ± 0.015 | 0.659 ± 0.013 | 0.601 ± 0.016 | 0.654 ± 0.014 | 0.584 ± 0.012 | 0.620 ± 0.019 | 0.503 ± 0.035 |
| | RGCN | 0.700 ± 0.013 | 0.675 ± 0.009 | 0.593 ± 0.030 | 0.536 ± 0.029 | 0.597 ± 0.028 | 0.530 ± 0.030 | 0.615 ± 0.014 | 0.500 ± 0.041 |
| | SM GDC ($T = 1.0$) | 0.699 ± 0.012 | 0.686 ± 0.012 | 0.664 ± 0.009 | 0.606 ± 0.012 | 0.660 ± 0.005 | 0.603 ± 0.006 | 0.617 ± 0.004 | 0.502 ± 0.033 |
| | SM GDC ($T = 0.5$) | **0.704 ± 0.011** | **0.694 ± 0.012** | 0.674 ± 0.009 | 0.631 ± 0.012 | 0.672 ± 0.009 | 0.636 ± 0.007 | 0.612 ± 0.006 | 0.506 ± 0.028 |
| | SM GDC ($T = 0.2$) | 0.687 ± 0.017 | 0.679 ± 0.017 | **0.682 ± 0.013** | **0.649 ± 0.012** | **0.678 ± 0.015** | **0.656 ± 0.015** | 0.613 ± 0.004 | 0.512 ± 0.013 |

Table 6: Summary of accumulated certifications and accuracy for the different architectures on Cora ML and Citeseer. We also report the accuracy of the base and smooth classifier (binary attr.).

| | Attr. A.&d. | A.&d. | Edges Add | Del. | Accuracy (base) | Accuracy (smooth) |
|---|---|---|---|---|---|---|
| **Cora ML [4]** | | | | | | |
| Vanilla GCN | $5.73 \pm 0.23$ | $1.84 \pm 0.01$ | $0.21 \pm 0.00$ | $4.42 \pm 0.01$ | $0.823 \pm 0.006$ | $0.816 \pm 0.006$ |
| Vanilla GDC | $5.80 \pm 0.06$ | $1.98 \pm 0.04$ | $0.20 \pm 0.00$ | $4.33 \pm 0.02$ | $\underline{0.825 \pm 0.007}$ | $0.824 \pm 0.007$ |
| Vanilla APPNP | $5.57 \pm 0.04$ | $3.37 \pm 0.02$ | $0.39 \pm 0.01$ | $4.61 \pm 0.00$ | $\mathbf{0.836 \pm 0.008}$ | $\mathbf{0.837 \pm 0.008}$ |
| Vanilla GAT | $\underline{5.83 \pm 0.09}$ | $1.26 \pm 0.09$ | $0.07 \pm 0.01$ | $4.03 \pm 0.07$ | $0.804 \pm 0.002$ | $0.807 \pm 0.004$ |
| SVD GCN | $5.51 \pm 0.14$ | $0.84 \pm 0.09$ | $0.08 \pm 0.02$ | $2.39 \pm 0.04$ | $0.772 \pm 0.008$ | $0.772 \pm 0.007$ |
| Jaccard GCN | $5.59 \pm 0.15$ | $0.86 \pm 0.10$ | $0.01 \pm 0.01$ | $4.39 \pm 0.00$ | $0.777 \pm 0.003$ | $0.778 \pm 0.003$ |
| RGCN | $4.64 \pm 0.07$ | $1.46 \pm 0.03$ | $0.12 \pm 0.01$ | $3.99 \pm 0.08$ | $0.796 \pm 0.007$ | $0.802 \pm 0.005$ |
| SM GCN ($T=50$) | $5.68 \pm 0.05$ | $1.86 \pm 0.03$ | $0.21 \pm 0.00$ | $4.44 \pm 0.02$ | $0.823 \pm 0.003$ | $\underline{0.825 \pm 0.003}$ |
| Dimmedian GDC | $4.66 \pm 0.05$ | $2.38 \pm 0.05$ | $0.32 \pm 0.01$ | $4.61 \pm 0.03$ | $0.804 \pm 0.002$ | $0.805 \pm 0.001$ |
| Medoid GDC | $1.98 \pm 0.07$ | $4.05 \pm 0.15$ | $0.51 \pm 0.02$ | $4.62 \pm 0.06$ | $0.742 \pm 0.008$ | $0.756 \pm 0.011$ |
| SM GDC ($T=1.0$) | $5.07 \pm 0.46$ | $4.31 \pm 0.68$ | $0.52 \pm 0.09$ | $4.71 \pm 0.08$ | $0.819 \pm 0.008$ | $0.822 \pm 0.007$ |
| SM GDC ($T=0.5$) | $4.15 \pm 0.67$ | $\underline{5.07 \pm 0.74}$ | $\underline{0.60 \pm 0.08}$ | $\underline{4.80 \pm 0.07}$ | $0.796 \pm 0.010$ | $0.803 \pm 0.008$ |
| SM GDC ($T=0.2$) | $2.90 \pm 0.95$ | $\mathbf{5.60 \pm 0.31}$ | $\mathbf{0.66 \pm 0.04}$ | $\mathbf{4.91 \pm 0.04}$ | $0.768 \pm 0.033$ | $0.775 \pm 0.034$ |
| SM GDC$^\dagger$ ($T=10$) | $\mathbf{7.15 \pm 0.01}$ | $1.12 \pm 0.06$ | $0.10 \pm 0.00$ | $1.63 \pm 0.01$ | $0.811 \pm 0.003$ | $0.814 \pm 0.002$ |
| **Citeseer [41]** | | | | | | |
| Vanilla GCN | $4.43 \pm 0.21$ | $1.24 \pm 0.10$ | $0.11 \pm 0.01$ | $3.88 \pm 0.17$ | $0.712 \pm 0.008$ | $0.712 \pm 0.009$ |
| Vanilla GDC | $\underline{5.21 \pm 0.22}$ | $1.13 \pm 0.10$ | $0.09 \pm 0.01$ | $3.85 \pm 0.13$ | $0.703 \pm 0.007$ | $0.701 \pm 0.007$ |
| Vanilla APPNP | $5.08 \pm 0.04$ | $2.21 \pm 0.06$ | $0.23 \pm 0.01$ | $4.16 \pm 0.04$ | $\underline{0.724 \pm 0.005}$ | $0.723 \pm 0.004$ |
| Vanilla GAT | $3.60 \pm 0.34$ | $0.66 \pm 0.13$ | $0.02 \pm 0.01$ | $3.24 \pm 0.48$ | $0.652 \pm 0.034$ | $0.634 \pm 0.044$ |
| SVD GCN | $3.46 \pm 0.13$ | $0.52 \pm 0.11$ | $0.00 \pm 0.00$ | $2.12 \pm 0.07$ | $0.638 \pm 0.015$ | $0.634 \pm 0.016$ |
| Jaccard GCN | $3.09 \pm 0.19$ | $1.42 \pm 0.10$ | $0.04 \pm 0.04$ | $3.96 \pm 0.14$ | $0.711 \pm 0.013$ | $0.712 \pm 0.012$ |
| RGCN | $4.27 \pm 0.18$ | $1.12 \pm 0.05$ | $0.09 \pm 0.01$ | $3.89 \pm 0.11$ | $0.719 \pm 0.012$ | $0.718 \pm 0.009$ |
| SM GCN ($T=50$) | $4.40 \pm 0.26$ | $1.25 \pm 0.10$ | $0.11 \pm 0.01$ | $3.90 \pm 0.17$ | $0.711 \pm 0.012$ | $0.710 \pm 0.013$ |
| Dimmedian GDC | $4.28 \pm 0.14$ | $1.42 \pm 0.05$ | $0.15 \pm 0.01$ | $3.92 \pm 0.08$ | $\mathbf{0.725 \pm 0.012}$ | $\mathbf{0.725 \pm 0.011}$ |
| Medoid GDC | $1.69 \pm 0.13$ | $2.41 \pm 0.04$ | $0.24 \pm 0.01$ | $3.97 \pm 0.06$ | $0.673 \pm 0.012$ | $0.689 \pm 0.007$ |
| SM GDC ($T=1.0$) | $4.93 \pm 0.24$ | $2.67 \pm 0.07$ | $0.32 \pm 0.02$ | $4.12 \pm 0.09$ | $0.711 \pm 0.010$ | $0.712 \pm 0.010$ |
| SM GDC ($T=0.5$) | $4.55 \pm 0.16$ | $\underline{3.62 \pm 0.19}$ | $0.48 \pm 0.03$ | $\underline{4.22 \pm 0.12}$ | $0.709 \pm 0.010$ | $0.716 \pm 0.010$ |
| SM GDC ($T=0.2$) | $3.52 \pm 0.17$ | $\mathbf{4.69 \pm 0.20}$ | $\mathbf{0.60 \pm 0.02}$ | $\mathbf{4.44 \pm 0.13}$ | $0.705 \pm 0.017$ | $0.714 \pm 0.014$ |
| SM GDC$^\dagger$ ($T=10$) | $\mathbf{5.62 \pm 0.15}$ | $0.17 \pm 0.02$ | $0.02 \pm 0.00$ | $0.82 \pm 0.12$ | $0.663 \pm 0.014$ | $0.654 \pm 0.014$ |
| **PubMed [47]** | | | | | | |
| Vanilla GCN | $\mathbf{4.40 \pm 0.20}$ | $3.23 \pm 0.17$ | $0.22 \pm 0.02$ | $4.19 \pm 0.06$ | $0.760 \pm 0.026$ | $0.744 \pm 0.031$ |
| Vanilla GDC | $\underline{4.32 \pm 0.11}$ | $3.10 \pm 0.04$ | $0.24 \pm 0.01$ | $4.05 \pm 0.15$ | $\mathbf{0.764 \pm 0.034}$ | $0.749 \pm 0.039$ |
| SM GDC ($T=1.0$) | $3.30 \pm 0.41$ | $5.13 \pm 0.66$ | $0.47 \pm 0.04$ | $4.24 \pm 0.16$ | $\underline{0.761 \pm 0.023}$ | $\mathbf{0.756 \pm 0.027}$ |
| SM GDC ($T=0.5$) | $3.10 \pm 0.60$ | $5.43 \pm 0.26$ | $0.56 \pm 0.03$ | $4.35 \pm 0.16$ | $0.751 \pm 0.015$ | $\underline{0.752 \pm 0.019}$ |
| SM GDC ($T=0.2$) | $2.44 \pm 0.40$ | $\mathbf{6.07 \pm 0.19}$ | $\mathbf{0.66 \pm 0.02}$ | $4.46 \pm 0.15$ | $0.729 \pm 0.014$ | $0.732 \pm 0.016$ |

## B.4 Certified robustness

Table 6 presents the complete results on CoraML, Citeseer as well as PubMed of our experiments with three-sigma error of the mean. On all three datasets Cora ML, Citeseer, and PubMed, our Soft Medoid GDC improves the robustness significantly. One of our Soft Medoid GDC models is in every structure robustness benchmark the most robust model. Moreover, we see that the the accuracy of the base classifier and the smooth classifier barely differ. We refer to § B.5 for our Soft Medoid$^\dagger$ GDC ($T = 10$) model, which improves the attribute robustness.

On Pubmed, due to the runtime, we do not report the results of the other defenses, select some important baselines. In their original papers, RGCN [58] is the only other defense [24, 52, 58] that reports results on a bigger data set, such as PubMed.

## B.5 Structural vs. attribute robustness

It is noticeable in Table 6 that increased robustness against structure attacks comes with a decreased robustness on attribute attacks (GCN as the baseline). This finding seems to be very consistent regardless of the chosen approach. For example, GAT outperforms APPNP on attribute attacks but lags behind APPNP on structure attacks.

For an increased robustness against attribute attacks, we came up with an alternative normalization

$$t_{\text{WSM}}^\dagger(\mathbf{X}, \mathbf{a}, T) = \left( \sum_{i=1}^{n} \mathbf{A}_{v,i} \right) \mathbf{s}^T \mathbf{X} \tag{24}$$

of our Soft Medoid estimator and different choice of temperature $T = 10$. Note that the Soft Medoid does not have the GCN as a special case Eq. 6 (§ 4). This configuration comes with the highest attribute robustness of all tested architectures (about 15% to 30% higher accumulated certifications w.r.t. attribute attacks than a GCN). For a comparison of the precise results see SM$^\dagger$ GDC ($T = 10$) in Table 6.

## B.6 Detailed comparison of certification ratios

Figure 8: (a) and (b) show the certification ratio over different radii for deletion $r_d$ and addition $r_a$, for a combined noise of $p_- = 0.4$ and $p_+ = 0.001$. (c) shows the case of only deleting edges ($p_- = 0.4$, $p_+ = 0$) and (d) only adding edges ($p_- = 0$, $p_+ = 0.001$). For each plot we set the contrary radius to zero (e.g. in (a) $r_a = 0$). We compare our Soft Medoid GDC against a GCN and the other defenses [24, 52, 58]. All plots are for Cora ML.

Figure 9: (a) shows the accumulated certifications over the degree (equal frequency binning), for a combined noise of $p_- = 0.4$ and $p_+ = 0.001$. (b) shows the case of only deleting edges ($p_- = 0.4$, $p_+ = 0$) and (c) only adding edges ($p_- = 0$, $p_+ = 0.001$). We compare our Soft Medoid GDC against a GCN and the other defenses [24, 52, 58]. All plots are for Cora ML.

We complement the certification rates with the accumulated certifications for different node degrees in Fig. 9 (compare to Fig. 5). Especially in the case of only adding edges we see the strength of our approach. For all the other approaches [24, 34, 36, 52, 58] we can basically certify 0 % of the low-degree nodes (degree $\leq 2$ before adding self-loops). In contrast with the Soft Medoid GDC, we are able to certify around 50% of the low-degree nodes!

In Fig. 8 we plot the certification ratios similarly to Fig. 6. We can see that the Soft Medoid GDC outperforms the other approaches by a large margin. Especially figures (b) and (d) highlight the unparalleled difficulty of adversarially added edges. Also in the other cases Fig. 8(c) only deleting and (d) only adding edges, the Soft Medoid GDC clearly outperforms the other architectures. The margin is especially large in the challenging case (d) of solely adding edges.