[Reviews · NeurIPS 2020]

Review 1

Summary and Contributions: This paper proposes a robust aggregration function, Soft Medoid, which is a differenitable generalization of the Medoid. The Soft Medoid function can be used to remarkably improve adversarial robustness of the graph neural networks. I think this is an interesting work.

Strengths: Based on the fact that the commonly used statistics in the aggregation functions of the graph neural network is not robust, e.g., average, sum, max, etc., then the authors of this paper proposed a novel soft-Medoid aggregation function, which enable robust statistics and differentiable. The soft-Medoid aggregation can be applied to improve adversarial robustness of GNNs. The author performed a decent amount of experiments to verify the advantage of using the proposed aggregation function. The paper is well-written and is a good fit to the NeurIPS community.

Weaknesses: How the robust certification is performed? More details are needed here. Line 72: ill-suited suited -> ill-suited

Correctness: The proposed method seems correct, and so is the empirical study.

Clarity: The paper is well written.

Relation to Prior Work: This paper did a great job in the literature review.

Reproducibility: Yes

Additional Feedback: Please address the Weaknesses section in the revision. --- Post-rebuttal After carefully read the paper, I fear the theoretical analysis and algorithms do not really match. This should be discussed.


Review 2

Summary and Contributions: The paper proposed a robust aggregation function, Soft Medoid, to replace the sum/mean operation used in conventional GCN. This operation is fully differentiable and yield better robustness compare with other defense methods.

Strengths: The method is novel and easy to follow. The theoretical analysis is comprehensive and convincing.

Weaknesses: I have some concerns about this paper: 1. It seems that the final Soft Medoid is just a softmax function with a controllable variable T in Eq.2. Although a plenty of proof and analysis are provided in the next sections, I still wonder whether it is enough to obtain robustness by this minor modification. 2. The assumption of the paper is too strong. They claimed that the reason for the vulnerability of conventional GNN is the aggregation function, such as a sum or mean, which might be distorted arbitrarily by a single outlier. This does not very make sense since the weights and activation functions also take charge of the GNN and may work as calibrator to these outliers. 3. The evaluation matrix is robustness certificates which is not straightforward enough. Why do not show the attack success rate directly from [3–7, 9, 12, 54]. 4. From Table 1, it seems like nature accuracy also decreases by the proposed Soft Medoid. What's the trade-off here compare with other defense methods? Minor concerns: The time cost is missing. The Cora ML and Citeseer are too small. Are there any results on large-scale datasets such as Pubmed? =================================== After read the rebutall, the author addressed my question well and I decided to rasie my score to 6.

Correctness: Maybe correct. I think it needs more experimental results to support the claims.

Clarity: The paper is well written.

Relation to Prior Work: Yes.

Reproducibility: Yes

Additional Feedback:


Review 3

Summary and Contributions: This paper proposed a new robust aggregation function for graph neural networks to defend against graph-structure based adversarial attacks, which can tolerate adversarial attacks with higher percentage of adversarial edges, meanwhile the function is fully differentiable and well suited for end-to-end deep learning. Experiments are conducted to justify the superior of the proposed aggregation method.

Strengths: 1. The proposed aggregation is simple and easy to understand with just one temperature hyper-parameter. 2. Robust analysis and theoretical certificate of the proposed aggregation on the breakdown point is given, which enjoys a larger value (0.5) compared to that (0) of sum and average etc. 3. The author conducted extensive experiments to validate the effect of soft Medoid, on different architectures and three different common used datasets, using hyper parameter searching and compared with different SOTA baselines against different structure based attacks, also consider graphs where node exhibits different degree distribution, hence I regard the results to be fairly convincing. 4. It's also pointed in the paper that the increasing defense against structure based attacks come at a cost of deceasing defense against attribute based adversarial attacks, some analysis is given. This paper helps to strength the robustness of GNN against structure based attacks by changing the aggregation function, which also gives some motivations on future research.

Weaknesses: 1. The time cost of the proposed aggregation function in practice is not given, though the worse case complexity is somehow analyzed, but practitioners may care more about the time, i.e., what’s the time cost to finish one epoch training/(or one inference) compared to the time of vanilla models like GCN? also what's the time cost compare to other defense? 2. Several previous attacks were used to evaluate the robustness of the aggregation, however, an defense-aware adversarial attack is not considered, i.e., knowing the change of the aggregation and its shortcomings, such as the larger bias for smaller perturbations, the adversary can potentially change the attack methodology to bypass the defense, in this attack and defense war, when a new defense is proposed, the author should consider an adaptive adversarial attack to make the defense strong and meaningful. 3. Though the (finite-sample) breakdown point certificate is given, but it's conditioned on when the result of the estimator can be arbitrarily placed, in practice, an adversary typically do not need to get the results arbitrarily place, but just to some degree, the authors should make efforts to defend this kind of adversary, also compare the proposed method against different baselines. 4. Three common used datasets (or graphs) are considered, but the size of them are basically in the same magnitude, the authors should consider larger graphs with more nodes, and report the defense result with computation cost compared to different baselines. 5. In line 158, it's mentioned that soft medoid comes with the risk of a higher bias for small perturbations and high epsilon, the author should take this effect into account when conducting experiments, to better show the shortcomings out to readers. 6. As found in the paper, this increased robustness against structure-based attacks comes with a cost of decreasing robustness on attribute attacks, the author should make it clear how much robustness lost to attribute attacks by using soft medoid aggregation, otherwise, the method just makes the model robust to structure attacks but super vulnerable to attribute inference attacks.

Correctness: Theoretical claims are justified by logical analysis, empirical methodology is also correct.

Clarity: Line 72, ill-suited suited, double suited

Relation to Prior Work: The author generalize the medoid function if robust statistics to a differentiable function soft medoid function, also the differences to previous defenses are analyzed and compared.

Reproducibility: Yes

Additional Feedback: I have read the author feedback, it addressed some of my concerns, such as time cost in Table B and defense results against several attacks in Table A, the method is straightforward and simple, proposed certificate is new and interesting, hence I will still vote accept for this paper. ----------------------------- Most are summarized above, one more advice is: can this aggregation method be applied in other areas to defend adversaries, such as in federated learning, when a artificially generated participant joins, can this aggregation makes the model robust to this method?

[Author Response · NeurIPS 2020]

**R2: Soft Medoid equation.** We apologize for any confusion about Eq. 2 and 3. We use the softmax with temperature only for approximating the $\arg\min$ operation of the Medoid (a multivariate generalization of the median). Thus, the temperature allows us to interpolate between the sample mean and the Medoid. In our proof, we show that the Soft Medoid (with a finite temperature) has the same breakdown point as the Medoid, which is well known to be robust.

**R2: Assumptions** It is not our intention to claim that the aggregation is the only reason for adversarial vulnerability of a GNN (will be clarified). We reason that on top of the usual (potentially non-robust) neural network components, GNNs introduce additional (typically non-robust) aggregations. In Sec. 5, we leave the GNN unchanged except for the aggregation function and it does substantially increase the robustness. Further countermeasures w.r.t. adversarial vulnerability are orthogonal to our approach. Note that our approach also helps in case of multiple "outliers" (see next).

**R4: Breakdown point/risk.** With an appropriate budget, the adversary can only perturb a subset of the aggregation inputs with the goal of crossing the decision boundary (i.e. a very small perturbation magnitude is unlikely to suffice). As long as the adversary only controls the minority of inputs, our robust estimator comes with a bounded error regardless of the attack characteristics (i.e. no attack can distort the aggregation result arbitrarily). According to Figure 2, the bias of the Soft Medoid is much lower than of the sample mean for a wide range of the perturbed fraction $\epsilon$ as well as different magnitudes of the perturbation (not only if arbitrarily far away). Moreover, the Soft Medoid comes with a guarantee on $\sup_{\tilde{\mathbf{X}}_\epsilon} \|t(\tilde{\mathbf{X}}_\epsilon) - t(\mathbf{X})\|$ (e.g. see line 599 in Appendix). We expect this guaranteed upper bound to be *slowly* increasing on the interval $\epsilon \in [0, \epsilon^*]$. In conclusion, our approach is more robust for reasonable $\epsilon$ and this naturally resonates with the desire of unnoticeable (adversarial) attacks.

**R1/R2/R4: Certification.** We use certified robustness because it provides guarantees (it is agnostic to the individual attacks used). Note that no attack (including adaptive attacks) can achieve a lower accuracy than certified (i.e. we evaluate the worst-case). We acknowledge that robustness certification on graphs is fairly new and introduces additional complexity to the setup. We built upon existing certification of [51] and will extend the explanation in our paper.

**R2: Further attacks.** As suggested, Table A extends Sec. 5.3 with three additional attacks [4-6]; our method outperforms all baselines in evasion attack settings (perturbed test data). We report the mean margin and failure rate of targeted evasion Nettack [4] (budget $\Delta = d - 1$), and the accuracy after evasion PGD [5] (transfer) as well as poisoning

Table A: Empirical results with three additional attacks. $\epsilon$ is the fraction of altered edges.

| | | Acc. | Cert. Edges | | | Nettack | | Acc. PGD & Metattack for $\epsilon$ | | | |
|---|---|---|---|---|---|---|---|---|---|---|---|
| | | | A.&d. | Add | Del. | Marg. | Fail. r. | 0.15 | 0.25 | 0.15 | 0.25 |
| Cora ML [46] | Vanilla GCN | 0.813 | 1.809 | 0.204 | 4.370 | -0.414 | 0.181 | 0.671 | 0.612 | 0.488 | 0.383 |
| | Vanilla GDC | **0.818** | 1.968 | 0.206 | 4.315 | -0.482 | 0.150 | 0.666 | 0.606 | 0.467 | 0.342 |
| | SVD GCN | 0.761 | 0.900 | 0.069 | 2.461 | 0.165 | 0.608 | 0.664 | 0.619 | **0.594** | **0.496** |
| | Jaccard GCN | 0.813 | 1.814 | 0.207 | 4.308 | -0.462 | 0.258 | 0.668 | 0.612 | 0.5 | 0.415 |
| | RGCN | 0.763 | 1.481 | 0.165 | 3.879 | 0.003 | 0.350 | 0.629 | 0.576 | 0.493 | 0.359 |
| | SM GDC ($T=1.0$) | 0.814 | 4.538 | 0.543 | 4.695 | 0.086 | 0.542 | 0.677 | 0.627 | 0.546 | 0.433 |
| | SM GDC ($T=0.5$) | 0.786 | 5.496 | 0.642 | 4.776 | 0.111 | 0.525 | 0.681 | 0.636 | 0.557 | 0.459 |
| | SM GDC ($T=0.2$) | 0.758 | **5.955** | **0.692** | **4.860** | **0.237** | **0.617** | **0.69** | **0.66** | 0.566 | 0.474 |
| Citeseer [47] | Vanilla GCN | 0.693 | 1.256 | 0.119 | 3.737 | -0.557 | 0.025 | 0.575 | 0.518 | 0.529 | 0.439 |
| | Vanilla GDC | 0.683 | 1.168 | 0.103 | 3.673 | -0.507 | 0.050 | 0.542 | 0.48 | 0.546 | 0.45 |
| | SVD GCN | 0.621 | 0.500 | 0.001 | 2.112 | -0.011 | 0.508 | 0.564 | 0.52 | **0.605** | **0.531** |
| | Jaccard GCN | 0.698 | 1.233 | 0.115 | 3.804 | -0.430 | 0.167 | 0.585 | 0.534 | 0.578 | 0.503 |
| | RGCN | 0.664 | 1.005 | 0.083 | 3.260 | -0.054 | 0.408 | 0.525 | 0.482 | 0.57 | 0.5 |
| | SM GDC ($T=1.0$) | **0.709** | 2.331 | 0.276 | 3.967 | -0.079 | 0.358 | 0.581 | 0.531 | 0.567 | 0.502 |
| | SM GDC ($T=0.5$) | 0.705 | 3.316 | 0.417 | 4.026 | 0.095 | 0.508 | 0.597 | 0.554 | 0.569 | 0.506 |
| | SM GDC ($T=0.2$) | 0.691 | **4.522** | **0.565** | **4.173** | **0.306** | **0.650** | **0.616** | **0.592** | 0.569 | 0.512 |

Metattack [6]. In the case of poisoning (perturbed training data) on Citesser, all defenses perform comparably except the slightly better performing SVD GCN. On Cora ML, we clearly outperform Jaccard GCN and RGCN.

**R2/R4: Accuracy vs. robustness.** Based on Table A, Table 1, and Figure 3, we see that there is a tradeoff between accuracy and robustness (consistent with e.g. *Tsipras et al. ICLR 2019. Robustness May Be at Odds with Accuracy.*). If we tune the hyperparameters for a comparable accuracy, our approach is consistently more robust w.r.t. structure perturbations than all the other defenses (the only exception is the Metattack poisoning attack). Note that recently we were able to improve our results with row-wise adjacency matrix normalization [45] (see first columns of Table A).

Table B: Average duration (time cost in ms) of one training epoch (over 200 epochs, preprocessing counts once). We report "-" for an OOM (DeepRobust impl.). We used one 2.20 GHz core and one 1080 Ti (11 Gb).

| | Cora ML [46] | | Citeseer [47] | | PubMed [46] | |
|---|---|---|---|---|---|---|
| GDC Prepr. | | ✓ | | ✓ | | ✓ |
| SM GCN | 41.2 | 210.9 | 36.6 | 154.1 | 86 | 497.8 |
| SVD GCN | 119.4 | 120.8 | 66.3 | 67.3 | - | - |
| Jaccard GCN | 19.1 | 147.8 | 11.2 | 118.0 | 84.9 | 585.4 |
| RGCN | 8.7 | 7.5 | 6.3 | 9.3 | - | - |
| Vanilla GCN | 5.1 | 7.1 | 4.7 | 7.8 | 6 | 66.1 |
| Vanilla GAT | 15.2 | 65.6 | 11.8 | 53.3 | 46.4 | 270.8 |

**R2/R4: Datasets.** In Table 4 (Appendix), we report the results on PubMed. PubMed is about 10 times bigger than Citeseer (see Table 2). None of the referenced attacks/defenses [3–7, 9, 12, 54] uses a larger dataset. Note that our approach scales (runtime/space) with $\mathcal{O}(n)$ (SVD GCN has space cmplx. $\mathcal{O}(n^2)$).

**R2/R4: Time cost.** The Soft Medoid is comparable to the defenses SVD GCN and Jaccard GCN (see Table B).

**R4: Attribute robustness and federated training.** Please see Section C.4 for attribute robustness. Further, we agree that federated learning is the way to go for future research.

[Meta-Review · NeurIPS 2020]

This paper proposed a new robust aggregation function for graph neural networks to defend against adversarial attacks. The questions raised by the reviewers have been addressed properly in the rebuttal. However, one of the reviewers found that the theoretical analysis provided in this paper does not really prove the "adversarial robustness" of the proposed aggregation function. More specifically, the analysis only shows that an attacker is harder to turn the aggregated results into +-\infty, while for adversarial robustness it is necessary to show "the aggregated results won't change significantly with small input perturbation". AC and other reviewers agree with this point but think this paper still has enough novelty and empirical contributions. We encourage the authors to address this concern in the revised version.